



# Osmium isotope and trace elements reveal melting of Chhota Shigri Glacier, western Himalaya, insensitive to anthropogenic emission residues

SARWAR NIZAM[1]*, INDRA SEKHAR SEN[1], TANUJ SHUKLA[1], DAVID SELBY[2,3]

[1]Department of Earth Sciences, Indian Institute of Technology Kanpur, Kanpur, UP 208016, India.
[2]Department of Earth Sciences, University of Durham, Durham DH1 3LE, UK.
[3]State Key Laboratory of Geological Processes and Mineral Resources, School of Earth Resources, China

*Correspondence to: sarwar@iitk.ac.in; isen@iitk.ac.in

## Abstract

The western Himalaya glaciers seasonally melt, in part, controlled by the presence of ice surface impurities in the form of dust, organic, and inorganic particles. The hitherto knowledge that dark-colored impurities on the ice surface are a mechanistic driver of heat absorption and thus enhancing ice mass wasting makes understanding the concentrations, origin, and pathways of emission residues on the glacier surface a global concern to conserve the Himalayan ice mass that provides water to more than one billion people. Yet, the source, origin, and pathways of metal impurities on the ice surface of Himalayan glaciers remain poorly constrained. Here, we present major and trace element geochemistry, rhenium-osmium (Re-Os) isotopes composition of cryoconite—a dark-colored aggregate of mineral and organic materials—on the ablation zone of the Chhota Shigri Glacier (CSG) considered as a benchmark glacier for process understanding in the western Himalaya. We find that the cryoconite possesses elemental ratios and crustal enrichment factor that reveal a predominant crustal source. Further, the $^{187}Os/^{188}Os$ composition in cryoconite varies from non-radiogenic (0.36) to radiogenic (1.31) compositions. Using a three-component isotope mixing model we show that the Os in cryoconite is dominantly derived from local rocks with negligible input from anthropogenic Os sources. Given that the CSG has limited debris cover (~ 3.4%) and the near absence of anthropogenically derived particles; our results suggests that dark-colored surficial deposits of anthropogenic dust particles are not one of the significant drivers of glacier melting in the western Himalaya, as observed elsewhere.





# 1. Introduction

Air pollutants in the southern slopes of the Himalayas—the highest mountain range on Earth—has reached high-altitude sparsely populated regions such as Khumbu (5079 m a.s.l.) in Nepal (Bonasoni et al., 2010) and Hanle (4520 m a.s.l.) in India (Babu et al., 2011). The pollutants emitted from the power, industrial, transport, and domestic sectors in the Indo-Gangetic Plains (Saikawa et al., 2019) are transported to the high-altitude Himalaya by the southwest Indian Summer Monsoon (Cristofanelli et al., 2014; Singh et al., 2020). Given that the presence of anthropogenic emission residues on the Himalaya is linked to driving climate change, enhanced glacier melting, and downstream water resources, the ambient air and its suspended particulate matter in high-altitude Himalayan sites have been intensely monitored over last few decades for greenhouse gases such as $CO_x$, $NO_x$, $SO_2$, BC, and $PM_{2.5}$ (Ran et al., 2014; Rupakheti et al., 2017; Shrestha et al., 2010; Stockwell et al., 2016).

Among the particulate impurities, various types of carbon such as organic carbon (OC) and black carbon (BC) are the major focus of current research (Kuchiki et al., 2015; Wang et al., 2019; Yan et al., 2019; Nizam et al., 2020). Studies have shown that the Himalayan ice record exhibits a factor of three increase in BC between 1975-2000 (Kaspari et al., 2011). Despite the rise in BC concentrations, the concentration of other inorganic impurities such as metals released from similar emission sources remains underexplored. Existing data mainly focuses on the source of pollutants $CO_x$, $NO_x$, $SO_2$, BC, and $PM_{2.5}$, that are mostly derived from receptor and chemical transport modeling, and assessment of existing emission inventories (Alvarado et al., 2018; Bonasoni et al., 2010; Rupakheti et al., 2018; Yarragunta et al., 2020). Although these models provide invaluable insights to the influence of anthropogenic emissions on the Himalayan cryosphere, the results exhibit disagreement due to the large uncertainties in emission inventories and meteorological parameters. Therefore, given that the Himalayas is neighbored by some of the world's largest emitters of anthropogenic particles, understanding the concentrations, origin, and pathways of other anthropogenically emitted particles, such as metals, is required to ultimately understand the driving mechanism(s) enhancing ice mass wastage and its impact on the downstream population. Understanding the emission source of metals that also emits $CO_x$, $NO_x$, $SO_2$, BC, and $PM_{2.5}$, will permit the independent evaluation of emission sources and support estimates studies based on transport and receptor models (Jeong et al., 2016; Yang et al., 2020).



Thus, to understand the contribution of emission sources to the Himalayan glaciers, here we utilize major and trace element geochemistry together with Re-Os isotope systematics of glacial surface impurities given that Re-Os isotopes are independent to the rate and magnitude of emission, biological or physiochemical fractionation during transport, complex orography and

meteorological parameters. The Os-isotope composition ($^{187}Os/^{188}Os$) of natural and anthropogenic materials only records the time-integrated fractionation of the Re/Os ratio in the sources. Due to the extremely long half-life of $^{187}Re$ (ca. 42 billion years), the present-day $^{187}Os/^{188}Os$ composition of natural and anthropogenic materials practically remains constant throughout the process of particle generation, transportation, and deposition. As a result, long-

lived radiogenic $^{187}Re/^{188}Os$-$^{187}Os/^{188}Os$ systematic is an emerging tool widely used in tracing anthropogenic sources in precipitation, snow, and ice in remote regions (Chen et al., 2009; Rauch et al., 2005; Rodushkin et al., 2007; Sen et al., 2013), as well as marine systems (Ownsworth et al., 2019; Sproson et al., 2020). As such, the objective of the study is to apply major and trace element geochemistry, and $^{187}Os/^{188}Os$ compositions of cryoconite and moraine samples to

constrain the metal composition across the ablation zone of the Chhota Shigri Glacier (CSG) in the western Himalaya. Further, we evaluate and discuss the sources of the metal impurities on the ablation zone of CSG in relation to the reported values for the Himalayan glaciers.

## 2. Materials and methods
### 2.1 Study area

The majority of the western Himalayan glaciers are up to 19% debris covered (Brun et al., 2019); in contrast, the CSG only has 3.4% of its surface area covered with debris (Vincent et al., 2013) and is therefore an excellent site to study long-distance emission inputs. The CSG (32.28° N, 77.58° E) is located in the Lahaul-Spiti valley, western Himalaya, India at an elevation between

4050 to 6263 m a.s.l. (Figure 1). The glacier is also considered as a benchmark glacier for process understanding in the western Himalaya (Pandey et al., 2017; Wagnon et al., 2007). The total axial length of the glacier is ca. 9 km and covers an area of ca.15.5 km$^2$. The equilibrium line of the CSG occurs at an altitude of ca. 4900 m a.s.l. (Wagnon et al., 2007). Presently, CSG has a total ice mass of ~$1.57\times10^{12}$ kg (after ice density 900 kg/m$^3$ and ice volume 1.74 km$^{-3}$) (Cogley, 2012;

Ramsankaran et al., 2018) and has an ice mass wasting rate of 0.50 m w.e.a$^{-1}$ since the end of the last century (Azam et al., 2019). The meltwater feeds the Chandra River, which ultimately joins



the Indus (Sindhu) river and provides water to an excess of 300 million people in the region of the Indus Basin.

The CSG is surrounded and underlain by the late Proterozoic to early Cambrian age High Himalayan Crystalline Sequence (HHCS), which comprise highly metamorphosed granite-gneiss,

schistose gneiss, augen gneiss, muscovite-biotite schist, quartzite, schist and low-grade metamorphic rocks (Kumar and Dobhal, 1997). Specifically, in the CSG catchment the glacial valley walls consist of mildly metamorphosed, intensely deformed fine-grained black slates, phyllites, and schists. The HHCS are often intruded by late Proterozoic to early Paleozoic mafic dykes, sills, and pegmatitic veins (Thakur and Patel, 2012; Thöni et al., 2012; Miller et al., 2001).

Meteorological data sets recorded within the glacier catchment divide the year into four seasons. These are, a warm summer monsoon (June to September), a cold winter (December to March), a pre-monsoon (April to May) and post-monsoon (October to November) period (Azam et al., 2014). Two principle atmospheric circulation patterns are prevalent over the CSG region; the southwest monsoon, which originates in the Arabian Sea and the Bay of Bengal during the

summer from July through to September, and the Western Disturbances that originate in the Mediterranean Sea during winter from January through to April (Bookhagen and Burbank, 2010). The majority of the annual precipitation across the CSG region originates from Western Disturbances (79%), with the remaining 21% being derived from monsoon precipitation (Azam et al., 2019). Noteworthy is that the western disturbance brings heavy snowfall to the CSG region

(e.g., average 6.3 mm w.e.d$^{-1}$ was recorded between Dec 2012–Jan 2013), with only 0.5 mm w.e. d$^{-1}$ average rainfall being derived from the southwest monsoon, resulting in the region being within the monsoon-arid transition zone (Azam et al., 2014). In addition to the two principle atmosphere circulation patterns, Hybrid Single Particle Lagrangian Integrated Trajectory Model (HYSPLIT) modeling results reveal 50% of the air mass that reaches the glaciers originates from the west,

within ~250 km of the study site, while the remaining air masses being more long-range in origin (Nizam et al., 2020).

## 2.2 Sample collection

Twenty samples of supraglacial cryoconite were collected from cryoconite holes across the entire ablation zone of the CSG between 4500 to 4930 m a.s.l. (Figure 1). The light grey to dark

black colored sediments was collected into Corning$^{®}$ 50 mL centrifuge tube using pre-cleaned





plastic scoops. In addition to the cryoconite samples, seven moraine debris samples of ca. 0.3-0.5 kg (cobbles to clay size) were also collected in polyethylene sterile Whirl-Pack® bags. All samples were kept frozen prior to analysis. To permit evaluation of anthropogenic emission particulates in the CSG our sample set also comprises samples of Gondwana and Tertiary coal from two major

coalfields of India (Jharia in the state of Jharkhand and Makum in the state of Assam), and diesel engine exhaust particulates obtained from engine-exhaust experiments.

**2.3 Engine exhaust experiment**

To obtain diesel engine exhaust particulates, engine experiments were performed in a contemporary common rail direct injection (CRDI) diesel engine (Tata; Safari DICOR 3.0 L)

coupled with an eddy current dynamometer. The experiments were conducted in the Engine Research Laboratory at the Indian Institute of Technology Kanpur, India by following published protocols (Agarwal et al., 2018). In brief, exhaust particulates were collected on a 47 mm quartz filter paper ($PM_{2.5}$) that was fitted on the exhaust tail type of the CRDI diesel engine. The operating conditions of the engine were optimized to those types of vehicles driven in India

(outlined in Agarwal et al., 2018). To avoid the possibility of any contamination with previous experiments, the walls of the photochemical chamber were thoroughly cleaned using ethanol and high purity water (18.2 M$\Omega$ cm) from a Millipore water purification system, and before sampling, zero air supply (containing less than 0.1 ppm of total hydrocarbon) was used to flush the photochemical chamber thoroughly.

**2.4 Sample preparation**

To remove any moisture, all cryoconite and moraine samples were first dried on a hot plate at ~70 °C. Cryoconite sediments size varies from fine granule (≤3 mm) to clay-sized fraction (≤0.002 mm). Moraine samples were sieved to obtain bulk (<3 mm) and fine (<63 μm) fractions for geochemical analysis. The fine fraction serves as a proxy of any wind-blown material (Pye,

1995; Shao, 2009). Several handpicked rock particles (3-60 mm) from each moraine sample were collected to represent the local lithology (hereafter referred to as local rock). Bulk cryoconite (20-50 g), moraine (10-15 g fine fraction, 100 g bulk), local rock (100 g), and coal (100 g) samples were grounded to a homogeneous powder (20 μm) in an agate mill.



### 2.4.1 Major and trace element analysis

Major elements for the cryoconite, fine, and bulk moraine size fractions were measured with a Wavelength Dispersive X-Ray Fluorescence spectrometer (Rigaku ZSX Primus II) at the Indian Institute of Technology, Kanpur, India. Oxide abundances were determined from sample

powders that were fused with lithium tetra borate mixed with lithium bromide ($Li_2B_4O_7$-$LiBO_2$-LMR) flux (1:10 ratio) in a Pt crucible at 1800 °C for 30 minutes to form fused glass beads. The reference standard SDTSD-4 (stream sediment) from Canadian Certified Reference Material Projects was used as a calibration standard. Loss on ignition (LOI) was determined by heating 1 g sample in an oven from 100 to 1000 °C. A combination of NIST SRM 2709a (San Joaquin soil)

and NIST SRM-8704 (Buffalo River sediment) from the USGS was used to assess the data quality. The measured values of the reference materials were within the uncertainty of the certified values (Table 1).

Trace metal and Rare Earth Element (REE) concentration analyses were performed at the Indian Institute of Technology Kanpur, India on an Agilent Triple Quadrupole Inductively

Coupled Plasma Mass Spectrometer (QQQ-ICP-MS). Approximately 100 mg of sampler powder was digested in pre-cleaned Teflon beakers at 150± 5 °C using a 2 mL mixture of HF: $HNO_3$ (4:1) for 48 hours. After digestion, the samples were dried and re-dissolved in aqua regia for 24 hours. To remove any organic materials, the cryoconite samples were further digested in 2 mL of ultra-clean $HNO_3$ for 24 hours at 130°C. The digested samples were slowly evaporated to dryness at

100 °C, and if required, acid digestion steps were repeated for incomplete digestion. Trace element concentrations were determined from ~350 ppm total dissolved solid solutions. Three procedural blanks, and three Reference Material GS-N (granite), GeoPT28 (shale) and NIST-2704 (river sediment) were also digested following the same procedures. All samples and standards were spiked with ~5 ppb Rh internal standard solution to monitor and correct any drift in intensities due

to matrix/drift effects. The instrument was run both in He and $O_2$ gas reaction mode to optimize the separation of measured isotopes from polyatomic interferences. Samples duplicates show reproducibility within 5% for most of the elements (Table 2). The final concentrations are all blank corrected using the average procedural blank concentrations. Average blank corrections were less than 1% for the majority of the elements. The measured trace element concentrations are in

agreement with the reference materials (Table 2 and 3).





### 2.4.2 Rhenium-Osmium (Re-Os) analysis

Ten cryoconite, two moraine samples (<63 μm fraction), four coal samples, and two engine exhaust particulates samples were selected for Re-Os analysis. The Re-Os concentration and isotopic compositions were determined at the Durham Geochemistry Centre using aqua regia

carius-tube digestion isotope-dilution negative ion mass spectrometry analytical protocols (Cumming et al., 2013; Selby et al., 2009). Approximately, 1 g of cryoconite and moraine, 0.2 g of coal, and 0.02 to 0.03 g of exhaust particulate were loaded into a carius tube with a known amount of mixed tracer solution (spike) of $^{190}$Os and $^{185}$Re and 9 ml of aqua regia solution. The carius tube was sealed and heated to 240 °C for 48 hours. The Os of the digested samples was

isolated and purified using standard solvent extraction (CHCl$_3$) and micro-distillation (CrO$_3$-H$_2$SO$_4$-HBr). The Re fraction was isolated and purified using NaOH-acetone solvent extraction and anion chromatography. The isolated Re and Os fractions were loaded onto Ni and Pt filament, respectively. The Re and Os isotopic compositions were determined using negative thermal ionization mass spectrometry using a Thermo Fisher TRITON mass spectrometer via static

Faraday collection mode for Re and ion-counting using a secondary electron multiplier in the peak-hopping mode for Os. Total procedural blanks were 2.1± 0.02 and 0.1± 0.01 ppt for Re and Os, respectively, with an average $^{187}$Os/$^{188}$Os value of 0.25± 0.03 (n = 2) for cryoconite and moraine analysis, and 2.3± 0.2 and 0.1± 0.02 ppt for Re and Os, respectively, with an average $^{187}$Os/$^{188}$Os value of 0.20± 0.06 (n = 4) for coal and exhaust particulate analysis. In-house standard solution

measurements yielded $^{185}$Re/$^{187}$Re value of 0.59786± 0.00014 (1 SD, n = 7) for the Re solution and a $^{187}$Os/$^{188}$Os value of 0.16085± 0.00017 (1 SD, n = 6) for the Durham Romil Osmium Solution (DROsS), which are in agreement with those reported in the previous studies (Percival et al., 2019). The average value plus uncertainty of the Re standard solution together with the natural $^{185}$Re/$^{187}$Re value of 0.5974 (Gramlich et al., 1973) is used for the Re sample mass fractionation correction.

Data reduction includes the instrumental mass fractionation, isobaric oxygen interference, and contribution of blanks and the tracer solution. The final two-sigma uncertainties of the Re-Os data include the fully propagated uncertainties of sample-spike weighing, tracer calibration, blank abundances, and isotope compositions, and the intermediate precision of the repeated measurements on the Re and Os reference solutions.

### 3. Results



### 3.1. Trace element systematics

Trace element concentrations of the cryoconite, bulk, and fine moraine were normalized to the local rock composition (Table 2; Figure 2). The fine moraine fractions and cryoconite exhibit similar patterns but possess higher trace element concentrations in comparison to the bulk moraine

fractions that resemble local rock compositions. The higher trace element concentrations in the fine moraine compared to bulk fraction are consistent with additional contamination/mixing or grain size/mineral sorting related enrichment as heavy metals (except Pb) including Sc, Ga, Sr, Nb, and Ta negatively correlate with $SiO_2$ (Pearson correlation coefficients R $\geq$-0.70 to -0.94, $p$ =.001-.05) as shown in Figure 3a and S1. (Cai et al., 2015; Thorpe et al., 2019). Additionally, an

overall negative correlation is exhibited between other elements (except Li, Be, Rb, Cs, Ta and Ba) against $SiO_2$ (Figure 3a). A similar relationship is exhibited between trace metals and $SiO_2$ in cryoconite samples (Figure 3b). In comparison to the fine moraine fraction, the cryoconite samples exhibit higher concentrations of V, Cr, Co, Ni, and Cd in more than 50% of the samples. Chondrite normalized REE concentrations show LREE enrichment and a negative Eu anomaly for

cryoconites and all moraine fractions (Figure S2).

The relationship of the cryoconite trace element ratios to the local rock composition suggests that the cryoconite composition has a local crustal provenance. For example, Cd/Zn (0.001-0.007) and Pb/Cu (0.84-1.98) ratios are similar to that of the local rock signature (Cd/Zn=0.001-0.014 and Pb/Cu=0.45-5.87) (Table 2). Further, REE ratios including, La/Ce,

La/Sm, La/Yb and La/Lu ratios of cryoconite that vary between 0.44-0.51, 30-57, 15-26, and 106-194, respectively, are similar to local rock values (0.37-0.48, 33-91, 15-41 and 113-317, respectively). Noteworthy, the La/Ce, La/Sm, La/Yb and La/Lu ratios of cryoconite and moraine are much lower than anthropogenic emission sources (La/Ce = 1.3-1.8, La/Sm = 19-28, La/Yb = 135-950, La/Lu = 5400-1000; Kitto et al., 1992; Olmez and Gordon, 1985). As are the Co/Cs ratios

in the cryoconite samples (0.25 and 2.85, average =1.12± 0.78, n=20, 1 S.D. versus >2.5 for anthropogenic emissions (Geagea et al., 2007)). Further, with the exception of Cr, Ni, and Cd, the element enrichment factors (EF) values of ≤1 support a predominant crustal provenance for the geochemistry of the cryoconite (Figure 4). The EF values for Cr (2.2), Ni (2.7) and Cd (2.6) in cryoconite show a detectable non-crustal input.

### 3.2. Re-Os systematics





Rhenium and Os concentrations of the cryoconite, moraine, coal, and vehicular exhaust samples are reported in Table 4. In Figure 5a, the Re (ppb) and Os (ppt) concentrations together with TOC (%) are shown for cryoconite and the fine moraine fraction. Overall, cryoconite sampled between 4700-4930 m a.s.l. possess higher average TOC (~1.7 %), Re (0.24-0.47 ppb), and Os

(37-104 ppt) in comparison to cryoconite sampled between 4500-4700 m a.s.l. (average TOC = 0.8 %; Re = 0.16-0.21 ppb; Os = 11-32 ppt). The elevated TOC, Re and Os concentrations in cryoconite above 4700 m a.s.l. also correspond to higher concentration of heavy metals (Figure 5b). For example, trace metals such as Cr, V, Co, Cu, and Zn follow the trends shown by Re and Os (Figure 5b) suggesting that these elements are probably controlled by common mineral/sources

(Chen et al., 2016). The absolute abundances of Re and Os show a moderate positive correlation (R= 0.58, $p$ =.1) (Figure 3b and S3), with a similar correlation exhibited with TOC (R= 0.71 and 0.53, respectively).

The $^{187}Os/^{188}Os$ composition in the cryoconite ranges between 0.38 and 1.31, with cryoconite between 4700-4930 m a.s.l. possessing $^{187}Os/^{188}Os$ compositions between 0.38 and

0.83. Below 4700 m a.s.l. the $^{187}Os/^{188}Os$ composition of cryoconites are more radiogenic (Figure 5). The $^{187}Os/^{188}Os$ compositions show a significant negative correlation with Os concentrations (R=-0.94, $p$ =.001; Figure 3b and S3). Further, the $^{187}Os/^{188}Os$ compositions correlate positively with major oxides such as $SiO_2$, $Na_2O$, and $K_2O$ (R = 0.71-0.89, $p$ =.001-.05). In contrast, heavy REE (HREE: Er to Lu), MgO, $Fe_2O_3$, MnO, and $TiO_2$ exhibit a significant negative correlation

with $^{187}Os/^{188}Os$ ratios (R= -0.81 to -0.93, $p$ =.001-.01; Figure S3 and S4).

The Re and Os concentrations of two moraine samples from ~4750 and 4550 m a.s.l are 0.14 and 0.21 ppb and 13 and 30 ppt, being similar to the average upper continental crust (UCC) composition (Re = ~0.20 ppb; Os = ~31 ppt) (Esser and Turekian, 1993; Peucker-Ehrenbrink and Jahn, 2001). The Re and Os concentrations in Gondwana coal range between 0.26 and 0.76 ppb

and 7 and 18 ppt, respectively. For the Tertiary coal samples, the Re and Os concentrations range between 0.47 and 0.53 ppb and 63 and 726 ppt, respectively. Unlike cryoconite, the Gondwana and Tertiary coal show limited variability and are characterized by radiogenic ($^{187}Os/^{188}Os$ =1.61-1.64) and unradiogenic osmium isotope compositions ($^{187}Os/^{188}Os$ = 0.14 and 0.21), respectively. The analysis of two engine exhausts yields Re and Os concentrations of 0.07 and 1.04 ppb and 2

and 6 ppt, respectively. The engine exhaust samples are characterized by an unradiogenic osmium





isotope composition ($^{187}$Os/$^{188}$Os=0.21-0.22), similar to catalytic converters (Poirier and Gariépy, 2005).

## 4. Discussion

### 4.1. The provenance of cryoconite from trace elements and osmium isotopes

Glaciers are sites of active physical erosion, with their continuous movement effectively powdering rock units in the glacial catchment that can be further eroded by wind action and deflation (Brown et al., 1996; Sharp et al., 1995; Tranter et al., 2002). This local freshly weathered rock that is subjected to deflation will overwhelm any long-range dust transport signal by covering the neighboring glacier. This is illustrated by both the major (Figure 6) and trace element

systematics (Figure 2) of the CSG cryoconite, which are derived from weakly weathered rocks of the glacial catchment. Most of the cryoconite (14 samples,) exhibit Co, Cr, Ni and Sc enrichment that indicate a detectable non crustal component. These samples occur mainly in the upper reaches (except first sample) of the glacier ablation zone i.e. above 4700 m a.s.l. These samples further show light deviation from the local moraine composition towards a more mafic rock composition

(Figure 7), which is further supported by the La-Sc-Th composition for sediments and rock from glacier catchment and its surrounding region (Figure 8). Although the higher concentration of heavy metals (Figure 4) can also be attributed to anthropogenic contributions, given that the air-mass back trajectory modeling clearly shows that 50% of the air mass that reaches the glaciers originates from the west, within ~250 km of the study site and with limited inputs from the Indo-

Gangetic Basin (Nizam et al., 2020), only minor input from anthropogenic sources can be considered. Lastly, the chondrite normalized REE signature of moraine, local country rocks from the Himalaya, river sediments from glacial catchments, cryoconite and dust collected from ice core and snow (Figure 9) show similar REE patterns, all exhibiting REE enrichment, which is similar to granitic or shale (PAAS)-like sources, which are commonly observed in the Higher and the

Lesser Himalayan rocks and sediment (Miller et al., 2001; Ranjan and Banerjee, 2009; Taylor and McLennan, 1985). The anthropogenic dust mostly exhibits fractionated (enriched) LREE patterns with smooth HREE enrichment, and often contains a strong positive Gd anomaly (Geagea et al., 2007; Hatje et al., 2016). The cryoconite show a large range in $^{187}$Os/$^{188}$Os that overlap with both natural and anthropogenic sources that do not exhibit singular $^{187}$Os/$^{188}$Os signatures (Figure 10).



Thus, given that the $^{187}$Os/$^{188}$Os composition of potential end-members overlap, deconvoluting the source of Os in the glacial debris is challenging.

The cryoconite unradiogenic $^{187}$Os/$^{188}$Os signature can be explained by both contributions from natural Os sources such as cosmic dust, volcanic aerosols, mafic and ultramafic rocks, as

well as anthropogenic Os from catalytic converters that are often recycled ($^{187}$Os/$^{188}$Os = ~0.38) (Poirier and Gariépy, 2005), fossil fuels, smelting of chromite, base-metal sulfide, and PGE ores and municipal solid waste incinerators (MSWIs). Contributions from volcanic aerosols and cosmic dust seem unlikely due to the absence of active volcanism in and around the Himalaya (Fitch TJ, 1970), and the extremely low (40,000± 20,000 t yr$^{-1}$; t=10$^6$ g) global cosmic dust flux (Love and

Brownlee, 1993) that can significantly affect the $^{187}$Os/$^{188}$Os of sediments in the highly active glacial ablation zones. Further, carbon characterization (Ramped Pyrolysis Oxidation, RPO, δ$^{13}$C, and $^{14}$C chronology) of total organic carbon (OC) in both cryoconite and the <63 μm fraction of moraine from the same glacier reveals that the cryoconites have negligible contributions from fossil fuel emission sources (Nizam et al., 2020). The RPO, δ$^{13}$C, and $^{14}$C data reveal that 98.3±

1.6% of the OC is sourced from local biomass sources, atmospheric organic matter, and glacial microbes, with only 1.7± 1.6% of the OC being sourced from petrogenetic sources. Contribution from mafic rocks is plausible as ultramafic dykes, sills, and pegmatitic veins (early Proterozoic-late Paleozoic age) common in HHCS and locally in the glacial catchment (Miller et al., 2001; Thakur and Patel, 2012; Thöni et al., 2012). Given that the unradiogenic $^{187}$Os/$^{188}$Os signature in

the cryoconite correlates with high $Fe_2O_3$-MgO and low $SiO_2$ concentrations (Figure S4), a mafic/ultramafic source rock input is likely as supported by the cryoconite trace element geochemistry (Figures 7 and 8). Therefore, from trace element systematics, Re and Os concentrations, and $^{187}$Os/$^{188}$Os compositions it can be concluded that the glacial debris contains felsic and mafic/ultramafic rock components, with essentially no input from anthropogenic

sources.

## 4.2. Evaluation of Natural and Anthropogenic Sourced Osmium

The relative contributions from various source end-members were quantified using a three-component mixing model using $^{187}$Os/$^{188}$Os and Os concentration as tracers. The first end-member has been chosen to represent local rocks, which is very similar in composition to the Higher

Himalayan Crystalline Sequence (Pierson-Wickmann et al., 2000) and eroding upper continental



crust (Peucker-Ehrenbrink and Jahn, 2001). We assign a 30.4 ppt Os and $^{187}$Os/$^{188}$Os value of ~1.48 (Table 4). For the second end-member, mafic-ultramafic rocks were selected having an Os concentration of 850 ppt and an $^{187}$Os/$^{188}$Os value of ~0.12 (Table 4; Data from Meisel et al., 2001, Sample Number: KH80-100, peridotite xenoliths). To explain the data distribution (Figure 10a), the third end-member should contain low Os concentration and intermediate $^{187}$Os/$^{188}$Os composition. This end-member could represent an Os-poor mineral such as aeolian quartz or granitoid or gneisses (Peucker-Ehrenbrink and Blum, 1998) and therefore we assign an Os concentration of 1 ppt and a$^{187}$Os /$^{188}$Os value of ~0.90. It is noteworthy that the $^{187}$Os/$^{188}$Os composition could be more radiogenic and Os concentrations can be significantly lower (Peucker-Ehrenbrink and Blum, 1998).

Keeping in mind that the three end-members of our mixing model should enclose all data points (Figure 11a), we performed our mixing model with the defined end-members as outlined above. Our mixing calculations suggest that the cryoconite $^{187}$Os/$^{188}$Os signature is derived from local rocks (67.4± 18.6%), with fractional contributions from an Os-poor mineral phase (29.6± 19.9%), and limited input from the mafic-ultramafic rocks (3.0± 2.8%, Table 4, Figure 11b). In general, the upper elevation of the glacier showed a greater Os contribution from mafic rock/mineral phase, which is consistent with the trace element systematics (Figure 5b).

We acknowledge that the choice of end-member compositions will change the end-member contributions, but we emphasize that the conclusion of the study will not change. Further, we did not include any anthropogenic sources because enrichment factors, trace and major elements, supports a predominant crustal provenance for the cryoconite. Moreover, this is supported by the fact that the glacier ablation zone is free of fossil fuel derived carbon (Nizam et al., 2020) and therefore we did not include coal and engine exhaust as suitable end-members.

We additionally emphasize that the metal enrichment in the cryoconites can also be modified (enriched) by microbial processes. Given the elevated TOC concentration observed in upper ablation zone of the CSG a higher level of microbial activity is implied (Anesio et al., 2009). The $\delta^{13}$C enrichment (-18.19‰) in cryoconite samples and its relationship with N enrichment and or depletion supports contributions from photo-autotrophic and heterotrophic micro-organisms that may have also modified predominantly heavy metal (Cr, V, Ni, and Co) signature (Nizam et al., 2020).

### 4.3 Local versus long-distance dust transport



We utilized the REE ratios to trace the long-range dust transport over the Himalaya. We assume that the REE ratios have retained the source signatures, or went through similar transportation and transformation changes and therefore behaved conservatively between the source and sink (Ferrat et al., 2011; Olmez and Gordon, 1985; Taylor and McLennan, 1985). Since

the magnitude of anthropogenically released REE particles are insignificant when compared to other trace and other heavy metal signatures, and exhibit variability based on their geographical location, REE ratios can be effectively used to track the source of dust particles (Geagea et al., 2008; Zhang and Liu, 2004). We show that the CSG cryoconites are characterized by a higher Mid REE enrichment (MREE, average: $Gd_{ch}/Yb_{ch}>2.4$) and higher negative Eu anomaly (Eu*<0.40),

which is different to that of the glacial debris in the Tibetan plateau, Central Himalaya, and Chinese mountains that show lower average negative Eu anomalies (>0.5) and a lower MREE enrichment (average: CSG $Gd_{ch}/Yb_{ch} \leq 1.75$) (Chang et al., 2000; Li et al., 2012, 2011). The REE profiles of the cryoconite samples and $Gd_{ch}/Yb_{ch}$ ratios are similar to sediments from the Thar Desert in the western parts of Indian sub-continent (Ferrat et al., 2012; Roy and Smykatz-Kloss, 2007). This is

further supported by other REE ratios such as $Nd_{ch}/Sm_{ch}$ vs $La_{ch}/Sm_{ch}$, and $Y_{ch}/Nd_{ch}$ vs $Eu_{ch}/Sm_{ch},$ which also shows small input from the Sahara (Figure 12). Further, previously we have shown that the 50% of the air mass (annual basis) originates far (>1000 km) from west of the receptor (CSG) site (Nizam et al., 2020). Therefore, on the basis of the REE systematics and air mass back trajectory models (Nizam et al., 2020) we conclude that the CSG receives limited long-range dust

inputs from the Thar deserts, which is located approximately 750 km northwest of the sampling site. It is noteworthy that our results are in agreement with previous findings that show that the Himalayan glaciers receive limited long-range transported dust from the Thar Desert. However, near-modern [14]C ages of cryoconite debris contradict the presence of long-range inputs from Thar Desert that are expected to carry an old [14]C signature (Nizam et al., 2020).

**4.4 Implications of absence of anthropogenic emission residues in the Western Himalayan glacier**

The Himalayan glaciers receive a large supply of metals in the form of mineral dust from wind-blown rock dust from the erosion of the upper continental crust and mafic rocks, with inputs from anthropogenic sources (Casey, 2012; Cristofanelli et al., 2014; Wake and Mayewski, 1993).

The knowledge of the amount, composition, and source of dust and anthropogenic emission is essential to calculate the heat-absorbing capacity of the residue, and in turn, ice melting rates. For





example, 1 ppb of BC residue on the glacier has the same effect on albedo as that of 75 ppb dust (Jacobi et al., 2015). In this study, we show that CSG has negligible metal impurity residues from anthropogenic sources. It is well established that glaciers in the CSG basin are losing mass at an average rate of 0.50 meter water-equivalent per year (m w.e. $yr^{-1}$) over the last two decades (Azam

et al., 2019), which is higher than the central (0.35 m w.e. $yr^{-1}$) and eastern Himalayan (0.43 m w.e. $yr^{-1}$) glaciers having higher debris covered glaciers (18-24%) and substantial contribution of anthropogenic sourced pollutants from the Indian subcontinent (Babu et al., 2011; Brun et al., 2017; Li et al., 2016). The regional differences between glaciers with impurities from anthropogenic sources (eastern and central Himalaya) vs. glaciers with non-anthropogenic

impurities (western Himalaya) can be best explained by air-mass transport pathways and precipitation regimes over the Himalaya. Our recent study reveals a marked difference between the rainfall amount between the western and eastern glacier sites with the former receiving lesser rainfall. In addition, the eastern and central Himalayan glacier sites are mostly influenced by air mass from the heavily polluted Indo-Gangetic Plains, whereas western Himalayan glacier are

dominated by air mass coming from the western parts of the Himalaya. (Nizam et al., 2020).  The near absence of anthropogenic particles on the nearly debris free CSG reveals that heat-absorbing anthropogenic particles deposited on the surface of the CSG are not one of the primary drivers behind CSG melting, as observed in other parts of the world such as the Greenland, Alaska, and Tibet (Dumont et al., 2014; Nagorski et al., 2019; Xu et al., 2009). Despite an increase in

anthropogenic emissions in the Indian subcontinent over the last 50 years (Crippa et al., 2018), we conclude that anthropogenic emission residues on the surface of CSG would not significantly enhance the glacier mass wastage rates in CSG in the near future.

## 5. Conclusions

The concentration, origin, and pathways of metals over the CSG—a benchmark glacier in

the western Himalaya—was investigated for the first time using major and trace elements, and Re-Os isotope systematics of supraglacial debris namely from cryoconite and moraine samples. Our study highlights two important points regarding the presence of metal impurities on the CSG glacier surface. Firstly, although the Himalaya is surrounded by some of the world's largest emitter of anthropogenic particles, we find limited anthropogenic metal impurities over the CSG likely

due to the source region of the air mass reaching the CSG. Mixing model calculations show that Os in the CSG is exclusively sourced from crustal rocks. Secondly, REE ratios and air-mass back



trajectory analysis reveal that the sediment/dust is mostly of local origin and only a minor fraction comes from far western parts of the Indian sub-continent, including the Thar Desert. We conclude that the enhanced CSG melting rate is insensitive to organic and inorganic anthropogenic emission deposits on the surface of the glacier.



**Data availability**

Full data are available as supplementary material.

**Author contributions**

I.S.S. conceived the study. S.N. performed laboratory measurements. S.N., I.S.S., T.S. and D.S.

analysed the data. S.N. and I.S.S. wrote the paper with input from all authors. I.S.S. handled

funding acquisition.

**Competing interests**

The authors declare that they have no competing interests.

**Acknowledgements**

This project is financially supported by Department of Science and Technology, Government of

India, Climate Change Program (SPLICE) Grant DST/CCP/Aerosol/86/2017(C) and Science &

Engineering Research Board (SERB) Grant (EMR/2015/000439) to I.S. Sen. S. Nizam is thankful

to Indian Institute of Technology-Kanpur (IIT-Kanpur) for Ph.D. scholarship. We thank IIT-

Kanpur and Durham University for providing access to instrumentation and support. We are

thankful to Mohd. Farooq Azam from IIT Indore for field sampling support. Sincere thanks to A.K.

Agarwal to provide access to the Engine Research Lab. Discussion with Michal Bizimis, Thomas

Meisel, and Soumita Boral greatly acknowledged. DS acknowledges the technical support of Chris

Ottley, Geoff Nowell, and Antonia Hofmann, and also the support of the TOTAL Endowment

Fund and the Dida Scholarship of CUG.

**Supplementary Materials:**

Figure S1-S4 and their captions.



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

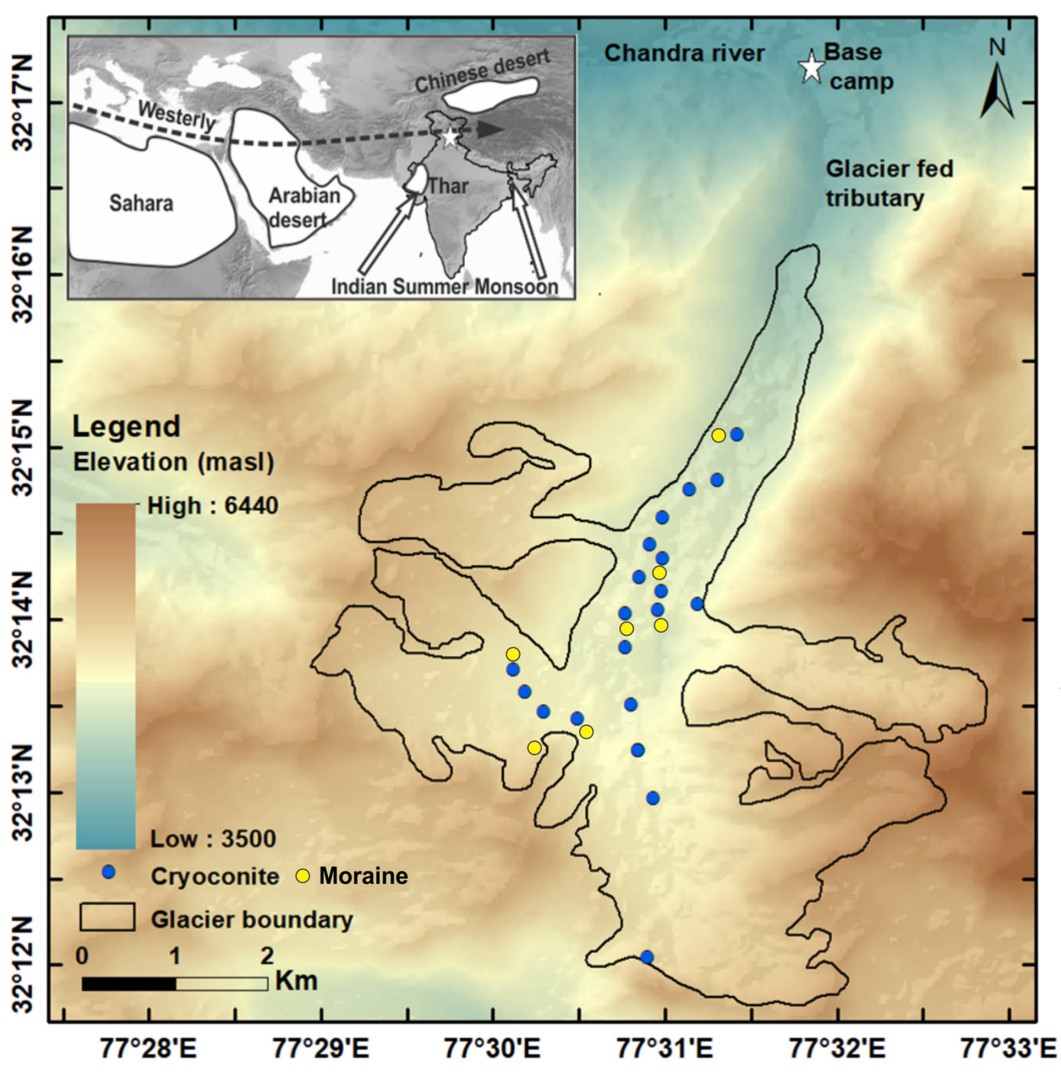

**Figure 1.** Map showing geographical location and cryoconite and moraine sampling points along the ablation zone of the CSG. Inset figure shows regional atmospheric circulation patterns in the glacier valley and desert distribution traced after (Parsons and Abrahams, 2009).

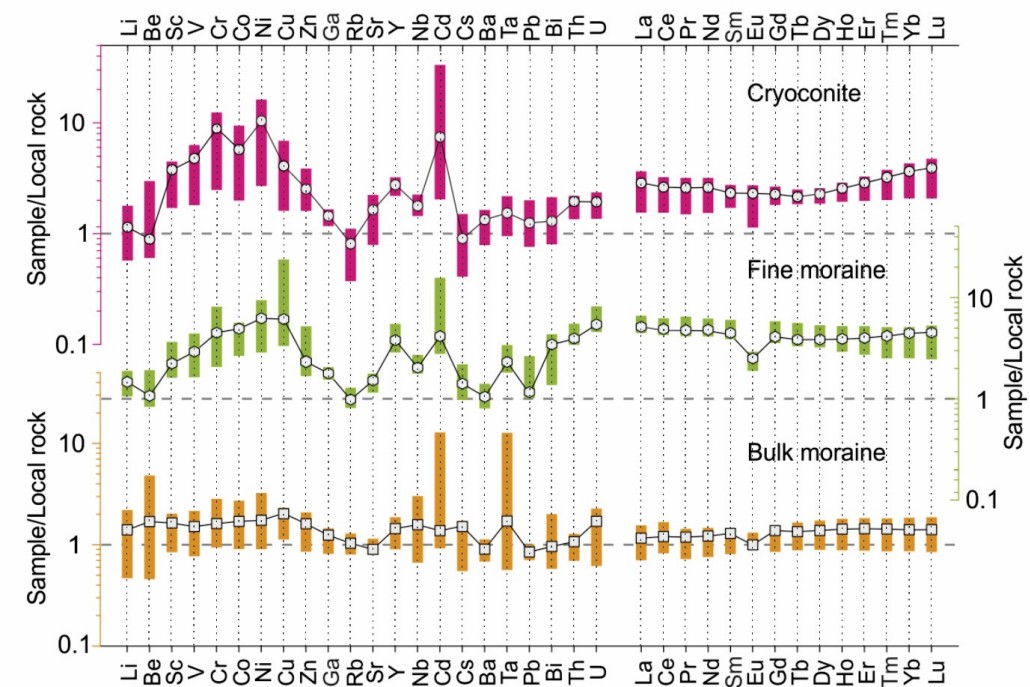

**Figure 2.** Local rock normalized elemental patterns for supraglacial cryoconite and moraine. Trace metal and rare earth elements are arranged in increasing atomic number. The upper and lower end of the bars represent the minimum and maximum values, the open symbol within the bar is the median value. Dashed straight line passing through unity is average local rock composition and any deviation from this line reflects additional source input or mineral sorting and elemental mobility attributed to physical weathering and transportation processes.





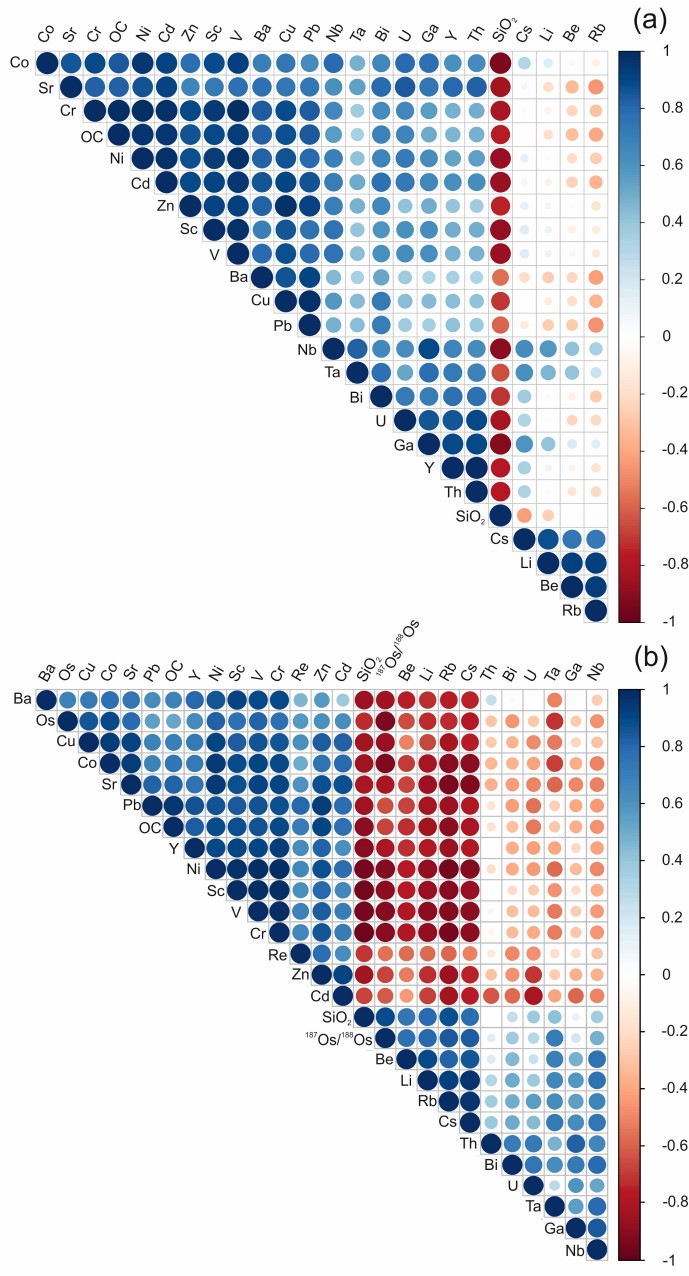

**Figure 3.** Correlogram showing Pearson correlation between trace metal, SiO2, Re-Os, and $^{187}Os/^{188}Os$ for (a) moraine (b) cryoconite. Positive correlations are displayed in blue and negative correlations in red color. The color intensity and the size of the circle are proportional to the correlation coefficients.

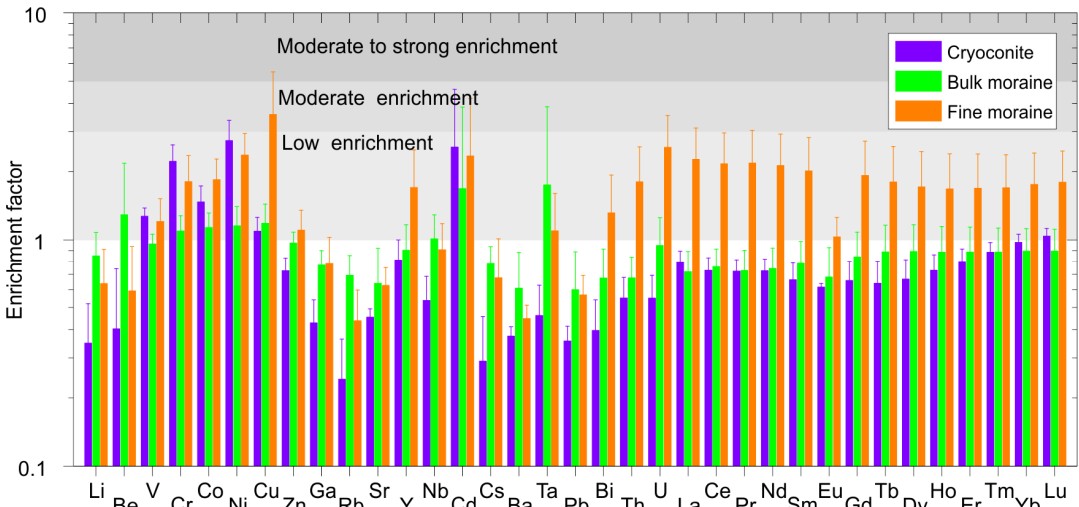

**Figure 4.** Enrichment Factor (EF) for trace metal and rare earth element in cryoconite (n=20, 1 SD), moraine (n=7, 1 SD) debris collected from CSG. EF (as $(X/Sc)_{sample}/(X/Sc)_{LR}$ for element X is calculated using Sc in local rock: LR (>3mm) (Barbieri, 2016).The EF value ≤1 corresponds to exclusively natural origin and EF value 1-3, 3-5, and 5-10 suggests low, moderate, and strong enrichment respectively (Barbieri, 2016).



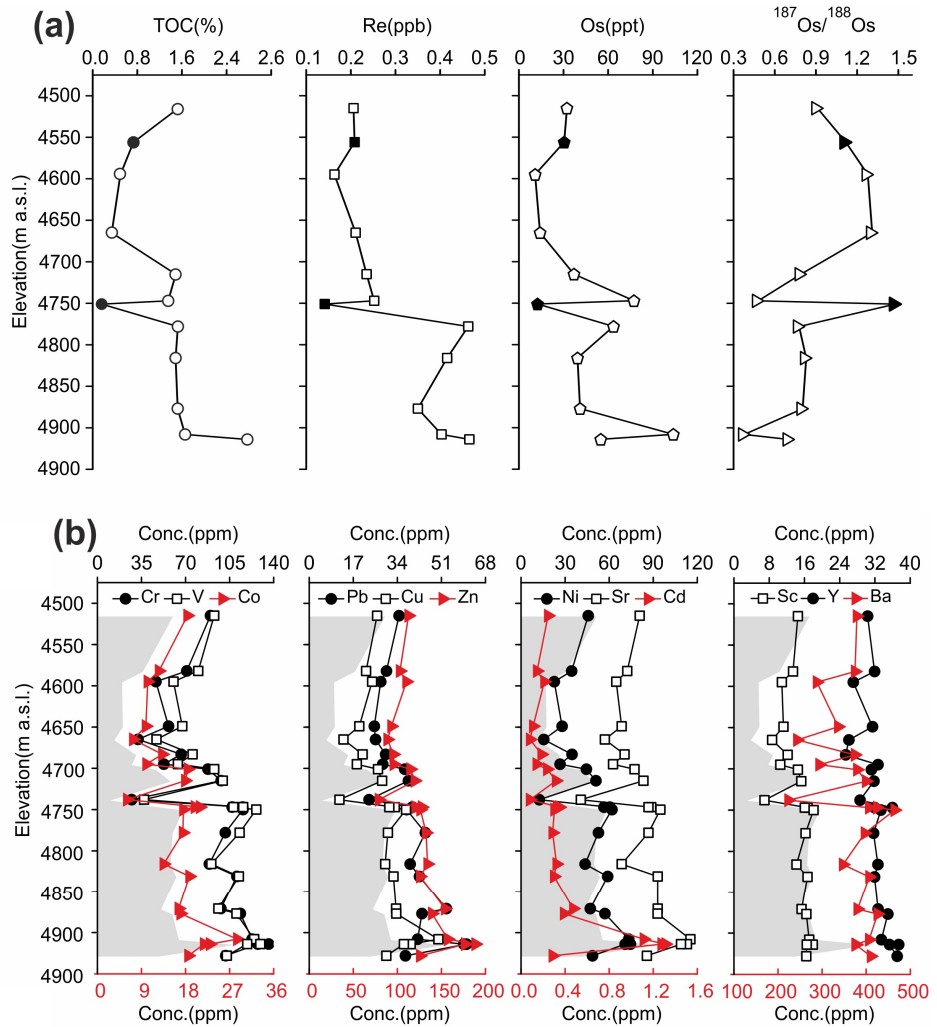

**Figure 5.** (a) Rhenium-Osmium concentration, [187]Os/[188]Os ratios and Total Organic Carbon (TOC) in cryoconite (open symbols) and fine moraine fraction (<63μm, filled symbols) (b) Trace metal concentration in cryoconite and TOC (scale is identical to a) along the ablation zone of the CSG. TOC data are from (Nizam et al., 2020).

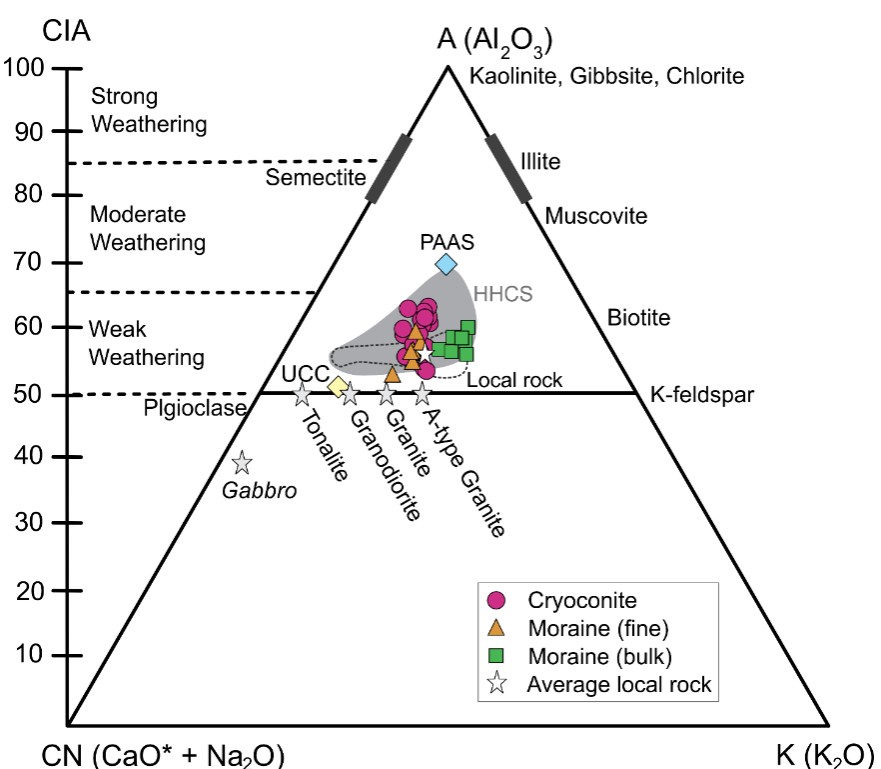

**Figure 6.** A-CN-K plot for supraglacial cryoconite, moraine, local granitoid rock, higher Himalayan Crystalline Sequence (HHCS), UCC, and PAAS. The vertical solid line plotted with a 10-unit interval represents weathering trends. Highly weathered sediments will move towards clay fraction end. The oxides abundance is in molar mass fraction and CaO* represents only the silicate fractions and determined by using the methodology described in an earlier study (Mclennan, 1993). Data source: Local rock (Maibam et al., 2016), HHCS (Richards et al., 2005), PAAS (Taylor and McLennan, 1985), and UCC (Rudnick and Gao, 2014).

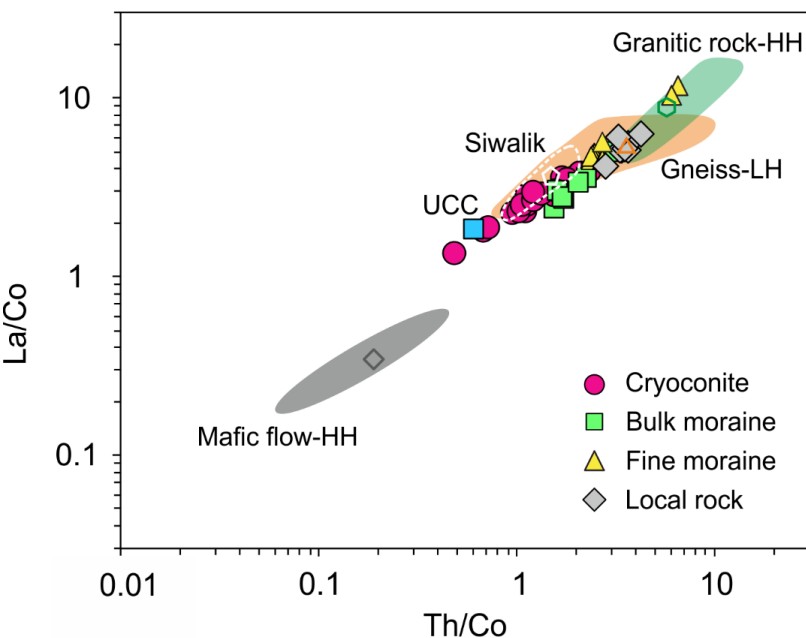

**Figure 7.** Bivariate concentration ratios plot of Th/Co versus La/Co showing cryoconite and moraine signature compared to different rock units and UCC. The open symbols are average values

5    of lithology plotted as field abbreviations LH and HH refer to Lower Himalaya and Higher Himalaya respectively. Data source: granitic and mafic rocks (Miller et al., 2001), gneiss (Islam et al., 2011), Siwalik: clastic rocks (Ranjan and Banerjee, 2009) and UCC (Rudnick and Gao, 2014).

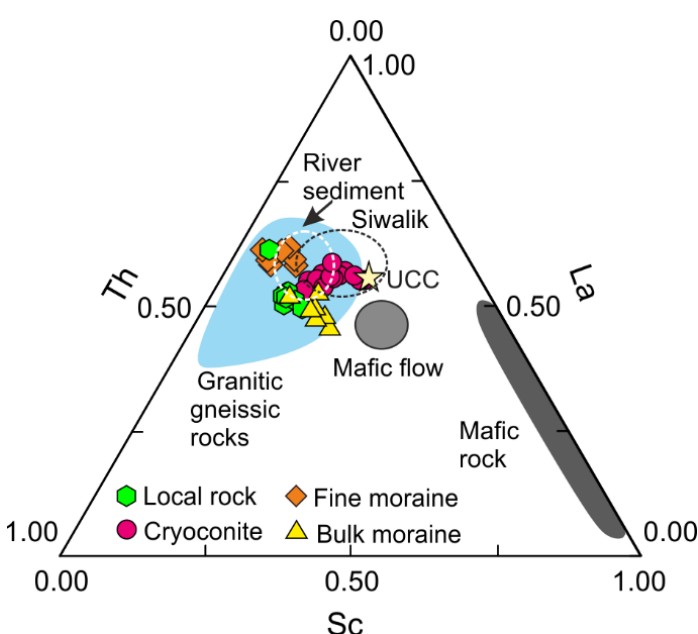

**Figure 8.** Ternary Th-Sc-La diagram illustrating lithological control on the composition of cryoconite and moraine debris. Fields of different rock types traced after published literature mentioned in Figure 7 except for mafic rocks (Srivastava and Sahai, 2001), and river sediments (Alizai et al., 2011).

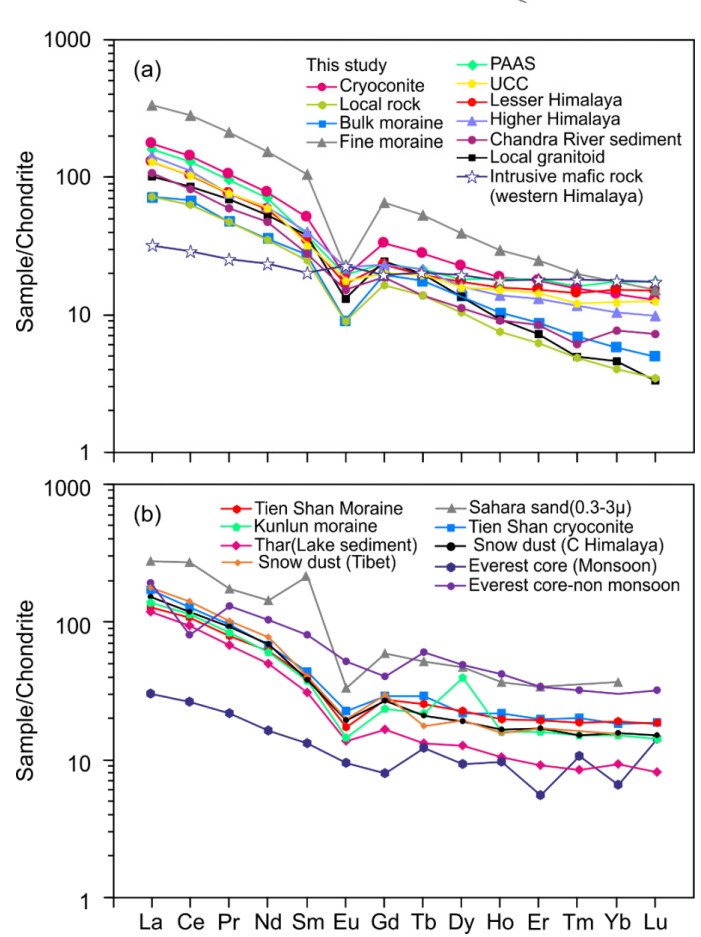

**Figure 9.** Comparison of chondrite normalized (McDonough and Sun, 1995) REE pattern of cryoconite and moraine with (a) local rock and sediment (b) snow dust, glacial moraine, sand and soil from other Himalayan regions and the Thar and Sahara deserts. Data source: (a) local granitoid (Maibam et al., 2016), Lesser Himalayan sediment (Das and Haake, 2003), Higher Himalaya (river sediment) (Panwar et al., 2017), PAAS (Taylor and McLennan, 1985), intrusive mafic rock (Srivastava and Samal, 2019) and UCC (Rudnick and Gao, 2014); (b) Chinese glacial moraine cryoconite (Chang et al., 2000; Li et al., 2011), Glacial dust from Tibet and Nepal (Li et al., 2012) and Everest ice core dust (Zhang et al., 2009), Thar (Ferrat et al., 2011; Roy and Smykatz-Kloss, 2007), and Sahara sand (Castillo et al., 2008).



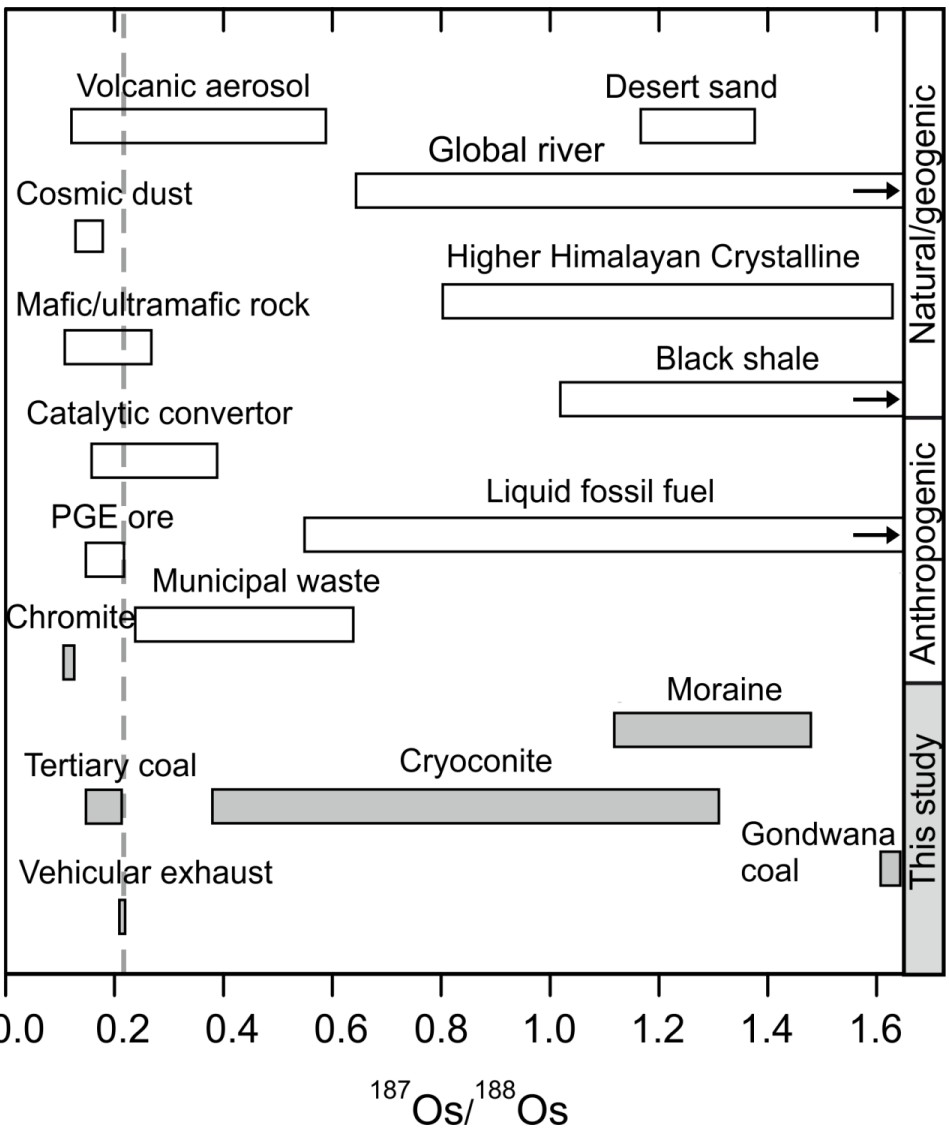

**Figure 10.** Comparison of osmium isotopic ratio of cryoconite and moraine with different potential sources. The dashed vertical line marks the limit of the measured unradiogenic Os that can be directly contributed by vehicular emission in the atmosphere. Moraine Os isotopic ratios lie within the crustal range of both eroding i.e. global river (0.64-2.94) (Levasseur et al., 1999) and non-eroding crust: Higher Himalayan Crystalline Sequence: 0.80-1.27 (Pierson-Wickmann et al., 2000), black shale: 0.69-20 (van Acken et al., 2019; Ackerman et al., 2019; Selby and Creaser, 2003; Singh et al., 1999). Cryoconite, in contrast, exhibits a range in $^{187}Os/^{188}Os$ ratios that



encompass a crustal signature to an unradiogenic signature similar to that of recycled catalytic converters (0.38) (Poirier and Gariépy, 2005), volcanic aerosol: 0.12-0.59 (Krähenbühl et al., 1992; Yudovskaya et al., 2008) and municipal waste: 0.21-0.70 (Funari et al., 2016). Data reference: cosmic dust: ~0.13 (Schmitz et al., 1997; Walker et al., 2002), mafic/ultramafic rock:

5   0.11-0.27 (Hanski et al., 2001; Meisel et al., 2001), Taklimakan desert and Kunlun moraine: 1.16-1.38 (Hattori et al., 2003), catalytic convertor: 0.16-0.38 (Poirier and Gariépy, 2005), liquid fossil fuel: 0.55-6 (Corrick et al., 2019; Cumming et al., 2014; Lillis and Selby, 2013; Selby et al., 2007), chromite ore: 0.104-0.128 (Mondal et al., 2007), PGE ore: 0.16-0.18 (Coggon et al., 2012), and municipal waste: 0.21-0.70 (Funari et al., 2016).

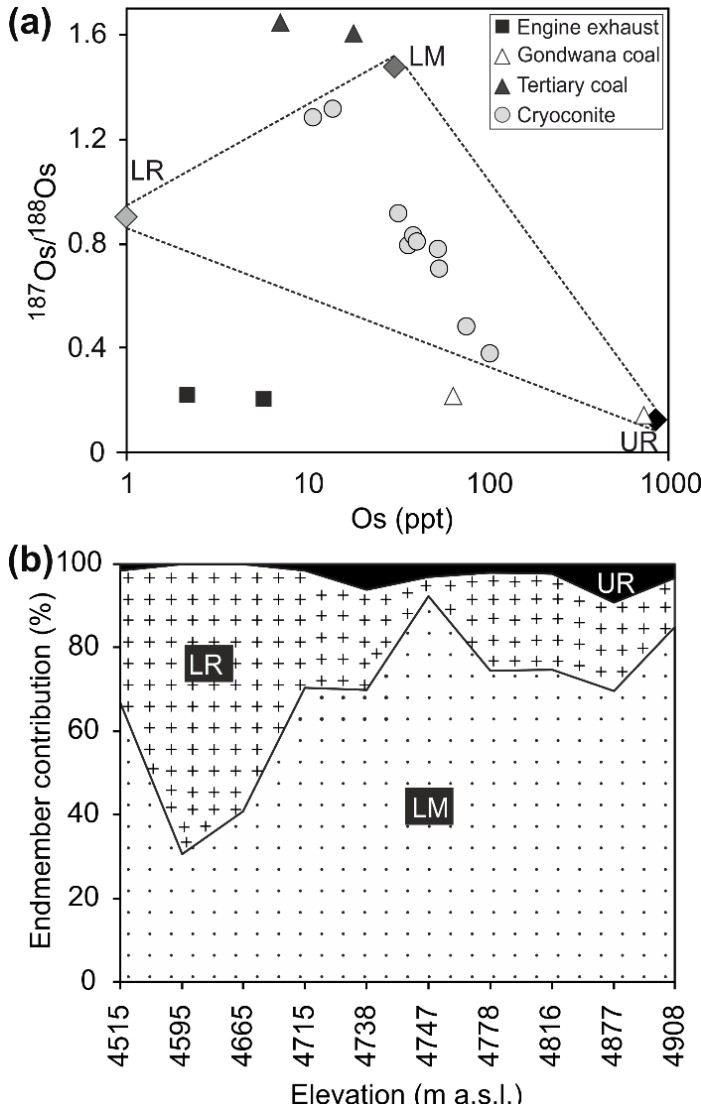

**Figure 11.** (a) Three-component mixing modeling plot for Os concentration and $^{187}Os/^{188}Os$ for cryoconite and the proposed three end-members: LM (Local Moraine), LR (Less Radiogenic Os poor mineral phase: aeolian quartz/granite/orthogneiss), and UR (Ultramafic Rocks). (b) Source percent contribution of the Os within the collected cryoconite relative to elevation along the ablation zone of the CSG. See text for discussion.


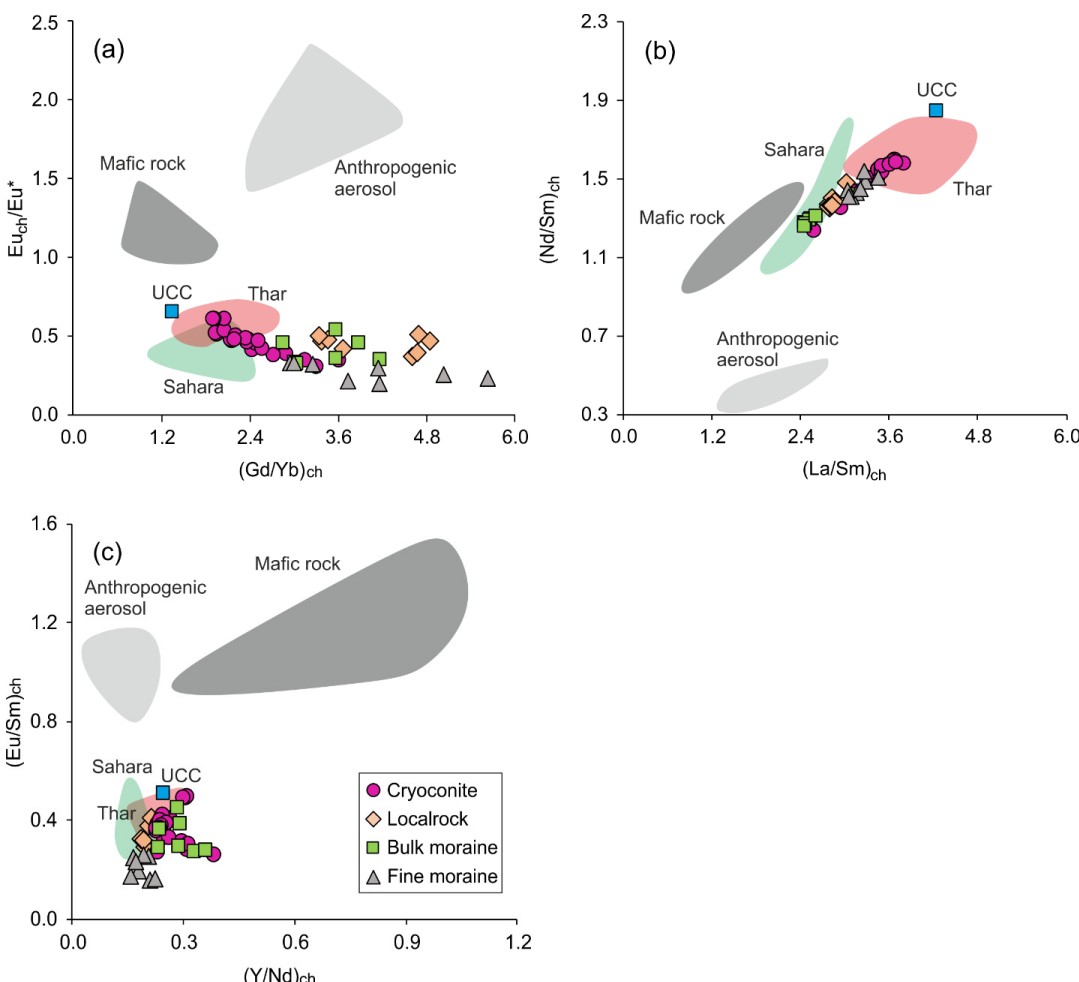

**Figure 12.** Source discrimination bivariate plots using chondrite normalized REE ratios: (a) $(Gd/Yb)_{ch}$ vs $Eu_{ch}/Eu^*$; (b) $(La/Sm)_{ch}$ vs $(Nd/Sm)_{ch}$ and (c) $(Y/Nd)_{ch}$ vs $Eu/Sm)_{ch}$ for glacial supraglacial cryoconite and moraine. Provenance data shown as fields are the same as Figure 9, except for the anthropogenic emission sources (Geagea et al., 2007; Sen et al., 2016).



**Table 1**. Major element abundance in cryoconite (Cx) and selected moraine samples (Fine fraction: DF:<63 µ, Bulk fraction: DB:<3 mm).

| Sample id | Elevation m a.s.l. | SiO$_2$ wt% | TiO$_2$ wt% | Al$_2$O$_3$ wt% | Fe$_2$O$_3$ wt% | MnO wt% | MgO wt% | CaO wt% | Na$_2$O wt% | K$_2$O wt% | P$_2$O$_5$ wt% | LOI wt% | Total wt% |
|---|---|---|---|---|---|---|---|---|---|---|---|---|---|
| C1 | 4515± 2 | 63.4 | 0.84 | 14.06 | 5.71 | 0.13 | 2.03 | 1.3 | 1.79 | 3.62 | 0.22 | 6.79 | 99.9 |
| C2 | 4582± 2 | 65.5 | 0.74 | 13.76 | 5.11 | 0.08 | 1.72 | 1.38 | 2.46 | 3.92 | 0.23 | 4.58 | 99.43 |
| C3 | 4595± 3 | 68.6 | 0.52 | 13.48 | 4.05 | 0.07 | 1.08 | 1.34 | 2.21 | 4.54 | 0.28 | 3.27 | 99.42 |
| C4 | 4649± 3 | 67.9 | 0.64 | 13.44 | 4.61 | 0.07 | 1.56 | 1.52 | 2.21 | 3.89 | 0.32 | 3.27 | 99.43 |
| C5 | 4665± 2 | 71.4 | 0.43 | 12.3 | 3.32 | 0.05 | 0.86 | 1.36 | 2.31 | 4.14 | 0.3 | 2 | 98.45 |
| C6 | 4683± 2 | 64.8 | 0.71 | 14.11 | 5.02 | 0.1 | 1.81 | 1.41 | 2.6 | 3.93 | 0.19 | 4.5 | 99.13 |
| C7 | 4695± 3 | 67.1 | 0.64 | 13.43 | 4.76 | 0.08 | 1.54 | 1.44 | 2.06 | 3.97 | 0.29 | 3.9 | 99.19 |
| C8 | 4701± 3 | 61.8 | 0.79 | 14.09 | 5.8 | 0.14 | 2.19 | 1.3 | 2.94 | 3.81 | 0.21 | 6.03 | 99.05 |
| C9 | 4715± 3 | 62.1 | 0.84 | 14.18 | 5.98 | 0.12 | 2.36 | 1.32 | 1.68 | 3.47 | 0.2 | 7.09 | 99.37 |
| C10 | 4738± 3 | 72.8 | 0.4 | 12.16 | 3.08 | 0.05 | 0.79 | 1.45 | 2.24 | 4.22 | 0.39 | 1.69 | 99.31 |
| C11 | 4747± 3 | 61 | 0.9 | 14.49 | 6.54 | 0.15 | 2.62 | 1.31 | 1.58 | 3.29 | 0.2 | 6.98 | 99.04 |
| C12 | 4750± 3 | 58.8 | 0.98 | 14.94 | 6.35 | 0.06 | 3.02 | 1.33 | 2.87 | 3.39 | 0.22 | 7.29 | 99.2 |
| C13 | 4778± 3 | 62 | 0.81 | 14.35 | 5.96 | 0.11 | 2.37 | 1.33 | 1.99 | 3.54 | 0.23 | 7.2 | 99.9 |
| C14 | 4816± 3 | 61 | 0.77 | 15.05 | 5.76 | 0.08 | 2.13 | 1.44 | 2.15 | 3.94 | 0.23 | 6.38 | 98.92 |
| C15 | 4831± 3 | 60.7 | 0.91 | 14.74 | 6.46 | 0.11 | 2.71 | 1.42 | 1.69 | 3.35 | 0.21 | 7.4 | 99.65 |
| C16 | 4871± 3 | 62.7 | 0.79 | 14.58 | 5.41 | 0.09 | 2.04 | 1.55 | 2.04 | 3.74 | 0.23 | 6.54 | 99.75 |
| C17 | 4877± 3 | 62.1 | 0.89 | 14.54 | 5.99 | 0.1 | 2.49 | 1.5 | 1.77 | 3.42 | 0.23 | 6.87 | 99.86 |
| C18 | 4908± 3 | 59.7 | 0.89 | 13.82 | 6.59 | 0.16 | 3.35 | 2 | 1.55 | 2.96 | 0.22 | 8.69 | 99.92 |
| C19 | 4914± 3 | 58.5 | 0.93 | 13.25 | 6.55 | 0.16 | 3.18 | 1.63 | 1.42 | 2.54 | 0.29 | 10.56 | 98.98 |
| C20 | 4928± 3 | 63.5 | 1.01 | 14.97 | 6.51 | 0.13 | 2.44 | 1.49 | 1.76 | 3.57 | 0.23 | 3.79 | 99.38 |
| DF6 | 4556± 4 | 63.2 | 0.88 | 15.1 | 6.04 | 0.08 | 2.04 | 1.57 | 2.27 | 3.76 | 0.3 | 4.48 | 99.75 |
| DF8 | 4657± 6 | 65.1 | 0.55 | 16.31 | 4.53 | 0.09 | 1.16 | 1.65 | 3.15 | 4.53 | 0.38 | 2.36 | 99.83 |
| DF10 | 4715± 3 | 63.6 | 0.73 | 15.09 | 5.54 | 0.11 | 1.7 | 1.51 | 2.45 | 4.22 | 0.27 | 3.54 | 98.79 |
| DF12 | 4751± 3 | 68.2 | 0.41 | 15.42 | 3.41 | 0.06 | 0.76 | 1.79 | 3.83 | 4.37 | 0.47 | 1.69 | 100.37 |
| DF14 | 4823± 2 | 63.4 | 0.52 | 17.29 | 4.57 | 0.09 | 1.06 | 1.53 | 3.45 | 4.85 | 0.35 | 2.85 | 99.96 |
| DF16 | 4890± 3 | 67.6 | 0.46 | 15.85 | 3.26 | 0.05 | 0.77 | 1.61 | 3.28 | 4.79 | 0.38 | 1.77 | 99.79 |
| DF18 | 4928± 2 | 66.1 | 0.73 | 14.31 | 5.11 | 0.09 | 1.51 | 1.58 | 2.54 | 3.99 | 0.32 | 3.28 | 99.53 |
| DB6 | 4556± 4 | 73.7 | 0.27 | 13.9 | 2.21 | 0.04 | 0.49 | 0.74 | 2.17 | 4.62 | 0.19 | 0.54 | 98.88 |
| DB8 | 4657± 6 | 70.7 | 0.47 | 13.7 | 3.89 | 0.07 | 0.81 | 0.82 | 1.93 | 5.02 | 0.26 | 1.38 | 99.04 |
| DB10 | 4715± 3 | 70.9 | 0.44 | 14.5 | 3.54 | 0.06 | 0.9 | 0.9 | 2.66 | 4.76 | 0.21 | 1.22 | 100.05 |
| DB12 | 4751± 3 | 72.3 | 0.35 | 13.49 | 2.86 | 0.04 | 0.59 | 0.78 | 2.37 | 4.84 | 0.24 | 1.07 | 98.96 |
| DB14 | 4823± 2 | 72.1 | 0.34 | 13.59 | 2.9 | 0.05 | 0.6 | 0.8 | 1.95 | 4.77 | 0.27 | 1.21 | 98.54 |
| DB16 | 4890± 3 | 77.4 | 0.17 | 11.59 | 1.36 | 0.02 | 0.26 | 0.71 | 1.81 | 4.61 | 0.24 | 0.61 | 98.74 |
| DB18 | 4928± 2 | 72.2 | 0.36 | 14.14 | 3.07 | 0.05 | 0.63 | 0.76 | 1.79 | 4.84 | 0.25 | 1.37 | 99.42 |
| **NIST reference material** | | | | | | | | | | | | | |
| SRM 2709a$^M$ | | 63.9 | 0.57 | 13.34 | 4.95 | 0.07 | 2.47 | 2.84 | 1.67 | 2.52 | 0.17 | 7.8 | 99.15 |
| | | ±0.73 | ±0.01 | ±0.82 | ±0.21 | ±0.00 | ±0.06 | ±0.24 | ±0.04 | ±0.03 | ±0.01 | ±0.01 | |
| SRM 2709a$^C$ | | 59.6-64 | 0.56-0.65 | 14.06-14.66 | 0.005-5.85 | 0.06-0.08 | 2.34-4.18 | 2.65-3.53 | 1.70-1.74 | 2.41-2.54 | 0.15-0.31 | | 93.61 |





| | | | | | | | | | | | | |
|---|---|---|---|---|---|---|---|---|---|---|---|---|
| SRM 8704[M] | 61.9 | 0.78 | 11.24 | 6.06 | 0.08 | 2.18 | 3.82± | 0.97 | 2.44 | 0.23 | 10.04 | 99.59 |
| | ±0.01 | ±0.01 | ±0.41 | ±0.12 | ±0.001 | ±0.09 | 0.08 | ±0.13 | ±0.01 | ±0.01 | ±0.01 | |
| SRM 8704[R] | 61.9- | 0.71- | 10.96- | 5.62- | 0.07- | 1.90- | 3.69- | 0.71- | 2.42- | 0.20- | | 99.53 |
| | 62.2 | 0.78 | 13.74 | 6.02 | 0.08 | 2.14 | 3.83 | 0.78 | 2.57 | 0.22 | | |

Note: Superscripts M refers to measured and C corresponds consensus values (http://georem.mpch-mainz.gwdg.de/sample_query.asp).





**Table 2.** Trace metal concentration (in ppm) in cryoconite and moraine samples. Cryoconite (Cx), local rocks (DB3-x, > 3 mm), moraine bulk fraction (DB-x, <3 mm), fine moraine (DF-x, <63 μm). Samples marked with apostrophe (') are duplicate analysis.

| Sample Element | C1 | C2 | C3 | C4 | C5 | C6 | C7 | C8 | C9 | C10 | C11 | GeoPT28[M] (n=3, 1 SD) | GeoPT28[C] |
|---|---|---|---|---|---|---|---|---|---|---|---|---|---|
| Li | 86 | 87 | 135 | 91 | 111 | 87 | 103 | 89 | 72 | 96 | 64 | 161± 17 | 164± 3 |
| Be | 3.9 | 5.7 | 12 | 8.5 | 6.5 | 9.7 | 6.5 | 4.3 | 3.2 | 9.8 | 3.1 | 3.29± 0.33 | 3.19± 0.08 |
| Sc | 15 | 13 | 11 | 11 | 8.7 | 12 | 10 | 15 | 15 | 6.9 | 17 | 24± 0.4 | 20± 0.2 |
| V | 93 | 80 | 60 | 67 | 47 | 76 | 64 | 93 | 100 | 37 | 116 | 250± 8 | 220± 1 |
| Cr | 90 | 71 | 46 | 56 | 32 | 66 | 53 | 88 | 98 | 27 | 115 | 117± 5 | 109± 1 |
| Co | 18 | 13 | 10 | 9.9 | 7.3 | 13 | 10 | 18 | 18 | 6.1 | 21 | 24± 0.8 | 23± 0.3 |
| Ni | 46 | 34 | 23 | 28 | 15 | 35 | 26 | 44 | 51 | 12 | 60 | 86± 3.1 | 83± 0.8 |
| Cu | 26 | 22 | 24 | 19 | 13 | 21 | 18 | 27 | 28 | 12 | 33 | 30± 1.2 | 31 ± 0.6 |
| Zn | 113 | 103 | 110 | 93 | 89 | 96 | 96 | 115 | 120 | 79 | 128 | 197± 7.1 | 187± 1.7 |
| Ga | 27 | 26 | 30 | 25 | 24 | 26 | 25 | 26 | 27 | 23 | 27 | 32 ± 0.8 | 27± 0.3 |
| Rb | 248 | 285 | 359 | 304 | 337 | 282 | 296 | 268 | 228 | 300 | 211 | 154± 6 | 147± 1.1 |
| Sr | 81 | 72 | 65 | 68 | 57 | 70 | 63 | 77 | 83 | 41 | 89 | 196± 4.7 | 178 ± 1.4 |
| Y | 30 | 32 | 27 | 31 | 26 | 25 | 33 | 31 | 32 | 29 | 36 | 36± 3.4 | 37± 0.3 |
| Nb | 25 | 24 | 30 | 23 | 22 | 25 | 23 | 24 | 22 | 19 | 23 | 19± 3.2 | 15 ± 0.2 |
| Cd | 0.25 | 0.14 | 0.21 | 0.11 | 0.08 | 0.19 | 0.14 | 0.24 | 0.32 | 0.08 | 0.35 | 0.4± 0.01 | 0.4± 0.19 |
| Cs | 17 | 20 | 29 | 22 | 24 | 18 | 23 | 22 | 16 | 24 | 13 | 7.9± 0.15 | 8.2± 0.1 |
| Ba | 379 | 375 | 288 | 337 | 243 | 374 | 294 | 380 | 400 | 224 | 407 | 772± 18 | 788± 8 |
| Ta | 2.7 | 2.4 | 3.4 | 2.3 | 2.4 | 3.4 | 2.5 | 2.2 | 1.9 | 1.9 | 2 | 2.4 ± 0.95 | 1.1± 0.03 |
| Pb | 35 | 30 | 28 | 25 | 26 | 29 | 28 | 37 | 38 | 23 | 40 | 31± 0.3 | 35± 0.3 |
| Bi | 1.38 | 1.1 | 1.3 | 0.82 | 0.57 | 0.76 | 0.75 | 0.97 | 0.85 | 0.77 | 0.81 | 0.59± 0.007 | 0.7± 0.02 |
| Th | 20 | 21 | 21 | 19 | 15 | 20 | 18 | 20 | 20 | 14 | 20 | 15± 0.04 | 16± 0.2 |
| U | 5.2 | 5.1 | 4.9 | 3.9 | 4.5 | 4.2 | 3.8 | 4.7 | 4.7 | 3.7 | 4.8 | 5.0± 0.12 | 5.76± 0.11 |

Note: Superscripts M and C correspond to measured and consensus values of GeoPT28 (http://georem.mpch-mainz.gwdg.de/sample_query.asp).




| Sample<br>Element | C11' | C 12 | C13 | C14 | C15 | C16 | C17 | C18 | C19 | C 19' | C20 | NIST-2704[M]<br>(n=3, 1 SD) | NIST-2704[C] |
|---|---|---|---|---|---|---|---|---|---|---|---|---|---|
| Li | 64 | 73 | 70 | 106 | 85 | 108 | 87 | 64 | 43 | 49 | 85 | 40± 4.8 | 48 |
| Be | 2.8 | 3.1 | 3.1 | 5.3 | 3.3 | 4.4 | 3.6 | 3.1 | 2.5 | 2.6 | 3.1 | 1.76± 0.2 | 0.97-1.5 |
| Sc | 16 | 18 | 16 | 14 | 17 | 15 | 16 | 17 | 18 | 17 | 16 | 14± 0.4 | 12 |
| V | 108 | 126 | 113 | 90 | 112 | 96 | 110 | 124 | 129 | 119 | 103 | 103± 4.9 | 95± 4 |
| Cr | 107 | 116 | 102 | 89 | 111 | 98 | 113 | 123 | 136 | 127 | 102 | 135± 5.6 | 135± 5 |
| Co | 20 | 18 | 17 | 14 | 19 | 17 | 17 | 29 | 23 | 22 | 19 | 14± 0.4 | 14± 0.6 |
| Ni | 56 | 62 | 53 | 44 | 59 | 47 | 57 | 73 | 74 | 71 | 49 | 43± 1.3 | 44± 3 |
| Cu | 31 | 37 | 30 | 29 | 33 | 33 | 34 | 50 | 39 | 37 | 30 | 92± 2.6 | 99± 5 |
| Zn | 121 | 126 | 133 | 134 | 126 | 154 | 139 | 157 | 189 | 177 | 126 | 391± 9 | 438± 12 |
| Ga | 25 | 28 | 28 | 30 | 27 | 29 | 27 | 25 | 23 | 21 | 30 | 18± 0.4 | 15 |
| Rb | 197 | 218 | 263 | 280 | 212 | 280 | 223 | 168 | 129 | 120 | 261 | 102± 0.4 | 100 |
| Sr | 87 | 95 | 87 | 68 | 93 | 93 | 93 | 115 | 113 | 109 | 86 | 133± 6.4 | 130 |
| Y | 32 | 33 | 32 | 33 | 32 | 33 | 35 | 33 | 37 | 35 | 37 | 29± 3.8 | 27-33 |
| Nb | 22 | 25 | 25 | 26 | 22 | 26 | 25 | 22 | 21 | 19 | 27 | 15± 1.3 | 15 |
| Cd | 0.32 | 0.3 | 0.29 | 0.32 | 0.3 | 0.47 | 0.39 | 1.11 | 1.31 | 1.28 | 0.29 | 3.1± 0.04 | 3.5±0.22 |
| Cs | 13 | 14 | 17 | 24 | 14 | 21 | 16 | 10 | 8.1 | 7.9 | 18 | 5.5± 0.1 | 6 |
| Ba | 425 | 465 | 397 | 348 | 407 | 381 | 426 | 407 | 377 | 376 | 411 | 398± 19 | 414± 12 |
| Ta | 1.9 | 2.2 | 2.6 | 2.9 | 1.8 | 2.4 | 2.4 | 1.5 | 2.5 | 2.4 | 2.6 | 1.2± 0.3 | 0.97 |
| Pb | 40 | 38 | 45 | 39 | 43 | 53 | 43 | 42 | 60 | 61 | 37 | 138± 3 | 161± 17 |
| Bi | 0.95 | 0.91 | 0.95 | 0.89 | 0.81 | 0.96 | 0.89 | 0.56 | 0.52 | 0.64 | 0.67 | 0.5± 0.01 | NA |
| Th | 20 | 20 | 22 | 21 | 20 | 20 | 22 | 14 | 15 | 16 | 22 | 7.8± 0.08 | 9.2 |
| U | 4.8 | 4.8 | 5.3 | 4.3 | 4.9 | 4.8 | 4.7 | 4 | 3.4 | 3.3 | 5.8 | 3.03± 0.04 | 3.13± 0.13 |

Note: Superscripts M and C correspond to average measured and consensus values of NIST-2704 (http://georem.mpch-mainz.gwdg.de/sample_query.asp).

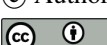



| Sample Element | DB3-6 | DB3-6' | DB3-8 | DB3-10 | DB3-12 | DB3-14 | DB3-16 | DB3-18 | *DB-6* | DB-8 | DB-10 | GS-N[M] (n=3, 1SD) | GS-N[C] |
|---|---|---|---|---|---|---|---|---|---|---|---|---|---|
| Li | 67 | 79 | 78 | 93 | 74 | 91 | 40 | 73 | *91* | 167 | 106 | 56± 7 | 56± 8 |
| Be | 5 | 5.2 | 4.1 | 4.3 | 2.9 | 28 | 2.1 | 3.8 | *7.1* | 14 | 6.6 | 5.7± 0.5 | 5.4± 0.15 |
| Sc | 3.5 | 3.6 | 1.8 | 6.3 | 4.5 | 4.5 | 3.5 | 4.8 | *5.4* | 8.2 | 8 | 9.1± 0.3 | 7.3± 0.7 |
| V | 21 | 20 | 24 | 29 | 18 | 18 | 17 | 22 | *28* | 44 | 44 | 71± 3 | 65± 17 |
| Cr | 11 | 11 | 14 | 17 | 10 | 10 | 10 | 13 | *17* | 25 | 31 | 60± 2 | 55± 11 |
| Co | 2.9 | 2.9 | 3.5 | 4.7 | 2.8 | 3.2 | 2.4 | 3.7 | *4* | 8.3 | 8 | 76± 6 | 65± 11 |
| Ni | 4.3 | 4.2 | 5.6 | 7 | 4.2 | 4.9 | 3.6 | 5.4 | *6.3* | 12 | 15 | 37± 1.3 | 34± 11 |
| Cu | 7.4 | 7.4 | 7.1 | 6.4 | 5.1 | 10 | 7 | 7.9 | *15* | 17 | 16 | 20± o.7 | 20± 1.8 |
| Zn | 47 | 48 | 56 | 59 | 50 | 44 | 42 | 63 | *72* | 102 | 83 | 49± 1.5 | 48± 9 |
| Ga | 16 | 17 | 20 | 21 | 19 | 18 | 16 | 21 | *21* | 27 | 23 | 28± 1.3 | 22± 6.6 |
| Rb | 317 | 330 | 308 | 340 | 346 | 310 | 272 | 345 | *335* | 418 | 315 | 208± 12 | 185± 14 |
| Sr | 50 | 51 | 51 | 65 | 52 | 47 | 58 | 58 | *48* | 44 | 59 | 648± 30 | 570± 52 |
| Y | 12 | 12 | 13 | 14 | 10 | 11 | 10 | 12 | *10* | 22 | 17 | 15± 1.8 | 16± 12 |
| Nb | 13 | 13 | 13 | 13 | 13 | 12 | 8.9 | 13 | *20* | 26 | 40 | 27± 3.1 | 21± 4.5 |
| Cd | 0.03 | 0.03 | 0.79 | 0.04 | 0.06 | 0.04 | 0.03 | 0.04 | *0.06* | 0.07 | 0.51 | 0.06±0.002 | 0.04±0.012 |
| Cs | 19 | 19 | 17 | 28 | 24 | 20 | 11 | 20 | *20* | 30 | 31 | 5.5± 0.26 | 5.4± 0.5 |
| Ba | 278 | 280 | 302 | 368 | 234 | 287 | 337 | 248 | *277* | 256 | 321 | 1436± 83 | 1400± 124 |
| Ta | 1.5 | 1.6 | 1.9 | 1.3 | 1.6 | 1.5 | 0.8 | 1.7 | *3.4* | 3.1 | 20 | 3.4± 0.85 | 2.6± 0.36 |
| Pb | 31 | 31 | 28 | 30 | 30 | 29 | 32 | 32 | *31* | 23 | 27 | 45± 2.2 | 53± 9.2 |
| Bi | 0.72 | 0.75 | 0.44 | 0.93 | 0.4 | 0.57 | 0.3 | 2.56 | *0.82* | 0.45 | 1.31 | 0.20-0.07 | 0.18-0.57 |
| Th | 9.7 | 9.7 | 11 | 17 | 10 | 9 | 10 | 13 | *7.1* | 13 | 13 | 40± 2.3 | 41± 6.9 |
| U | 1.6 | 1.6 | 2.9 | 2.6 | 2.2 | 3.2 | 2.2 | 6.1 | *1.5* | 4.2 | 3 | 7.1± 0.42 | 7.5± 1.71 |

Note: Superscripts M and C correspond to average measured and consensus values of GS-N (http://georem.mpch-mainz.gwdg.de/sample_query.asp).




| Sample Element | DB-12 | DB-14 | DB-16 | DB-18 | DF-6 | DF-6' | DF-8 | DF-10 | DF-12 | DF-14 | DF-16 | DF-18 |
|---|---|---|---|---|---|---|---|---|---|---|---|---|
| Li | 130 | 129 | 35 | 85 | 82 | 100 | 128 | 143 | 122 | 138 | 80 | 84 |
| Be | 11 | 20 | 1.9 | 2.4 | 4.2 | 4 | 7.8 | 4.8 | 8.1 | 7.5 | 4 | 3.5 |
| Sc | 6.5 | 6.6 | 3.4 | 7.6 | 15 | 14 | 8.9 | 9.2 | 6.6 | 8 | 8.5 | 11 |
| V | 31 | 31 | 16 | 30 | 91 | 88 | 52 | 89 | 33 | 43 | 39 | 68 |
| Cr | 17 | 18 | 10 | 18 | 89 | 90 | 37 | 62 | 23 | 26 | 29 | 67 |
| Co | 5 | 6.4 | 2.8 | 5.2 | 17 | 16 | 15 | 15 | 8.6 | 15 | 8.2 | 15 |
| Ni | 7.1 | 8.8 | 4.2 | 8 | 43 | 42 | 24 | 35 | 13 | 21 | 13 | 33 |
| Cu | 11 | 17 | 8.2 | 11 | 170 | 166 | 42 | 46 | 34 | 50 | 24 | 42 |
| Zn | 79 | 80 | 42 | 70 | 251 | 246 | 107 | 122 | 85 | 117 | 82 | 110 |
| Ga | 24 | 25 | 15 | 21 | 32 | 31 | 34 | 33 | 35 | 36 | 33 | 32 |
| Rb | 399 | 380 | 259 | 313 | 280 | 262 | 351 | 322 | 334 | 415 | 314 | 311 |
| Sr | 46 | 44 | 54 | 42 | 91 | 88 | 75 | 82 | 72 | 59 | 71 | 81 |
| Y | 17 | 21 | 13 | 13 | 47 | 45 | 39 | 36 | 64 | 33 | 57 | 43 |
| Nb | 21 | 21 | 8.7 | 16 | 30 | 29 | 27 | 26 | 26 | 35 | 24 | 25 |
| Cd | 0.04 | 0.05 | 0.04 | 0.05 | 0.28 | 0.28 | 0.14 | 0.61 | 0.11 | 0.12 | 0.11 | 0.19 |
| Cs | 32 | 30 | 11 | 21 | 22 | 22 | 32 | 43 | 35 | 42 | 19 | 24 |
| Ba | 234 | 216 | 280 | 194 | 399 | 399 | 287 | 357 | 242 | 229 | 269 | 308 |
| Ta | 2.7 | 2.6 | 0.9 | 1.2 | 4.1 | 3.6 | 3.3 | 3.6 | 5.1 | 5.3 | 3.3 | 2.8 |
| Pb | 26 | 25 | 30 | 21 | 76 | 80 | 31 | 37 | 33 | 31 | 32 | 35 |
| Bi | 0.62 | 0.62 | 0.38 | 0.64 | 2.6 | 2.81 | 1.36 | 2.6 | 2.67 | 1.85 | 0.88 | 1.46 |
| Th | 11 | 11 | 8.8 | 11 | 39 | 40 | 37 | 36 | 56 | 40 | 49 | 40 |
| U | 4.4 | 4.7 | 2.4 | 5.6 | 13 | 13 | 13 | 11 | 16 | 18 | 12 | 20 |



**Table 3.** Rare earth element concentration (in ppm) in cryoconite and moraine samples. Sample notions are same as Table 1 and Table 2 and apostrophe symbol refers to duplicate samples.

| Sample \ Element | La | Ce | Pr | Nd | Sm | Eu | Gd | Tb | Dy | Ho | Er | Tm | Yb | Lu |
|---|---|---|---|---|---|---|---|---|---|---|---|---|---|---|
| C1 | 46 | 96 | 10.6 | 38.3 | 7.82 | 1.141 | 6.8 | 1.016 | 5.61 | 1.051 | 2.84 | 0.385 | 2.31 | 0.324 |
| C2 | 45 | 93 | 10.1 | 37.3 | 8.03 | 1.004 | 6.97 | 1.051 | 5.72 | 1.062 | 2.88 | 0.384 | 2.3 | 0.316 |
| C3 | 39 | 85 | 9.34 | 34.2 | 7.73 | 0.806 | 6.75 | 1.007 | 5.17 | 0.871 | 2.19 | 0.267 | 1.5 | 0.202 |
| C4 | 35 | 76 | 8.54 | 31 | 7 | 0.837 | 6.32 | 0.98 | 5.12 | 0.924 | 2.39 | 0.308 | 1.76 | 0.237 |
| C5 | 28 | 59 | 6.67 | 24.6 | 5.9 | 0.638 | 5.4 | 0.877 | 4.63 | 0.795 | 2 | 0.248 | 1.38 | 0.184 |
| C6 | 38 | 83 | 8.95 | 32.6 | 6.8 | 0.932 | 5.82 | 0.863 | 4.7 | 0.86 | 2.32 | 0.316 | 1.89 | 0.265 |
| C7 | 35 | 78 | 8.3 | 30.5 | 6.89 | 0.8 | 6.17 | 0.97 | 5.2 | 0.935 | 2.43 | 0.315 | 1.83 | 0.249 |
| C8 | 43 | 92 | 10.1 | 36.6 | 7.69 | 1.084 | 6.57 | 1.006 | 5.61 | 1.049 | 2.93 | 0.404 | 2.46 | 0.336 |
| C9 | 45 | 96 | 10.4 | 38.4 | 7.97 | 1.142 | 6.77 | 1.016 | 5.74 | 1.073 | 2.98 | 0.417 | 2.54 | 0.357 |
| C10 | 24 | 54 | 5.82 | 21.8 | 5.73 | 0.565 | 5.67 | 0.952 | 4.95 | 0.837 | 2.06 | 0.253 | 1.38 | 0.179 |
| C11 | 47 | 97 | 10.8 | 39.2 | 7.96 | 1.238 | 6.93 | 1.065 | 6.11 | 1.189 | 3.32 | 0.465 | 2.84 | 0.408 |
| C 11' | 46 | 96 | 10.6 | 38.7 | 7.98 | 1.249 | 6.92 | 1.058 | 6.04 | 1.175 | 3.32 | 0.462 | 2.86 | 0.402 |
| C 12 | 50 | 101 | 10.9 | 39.9 | 8.18 | 1.32 | 7.04 | 1.045 | 5.88 | 1.16 | 3.24 | 0.448 | 2.75 | 0.389 |
| C 13 | 48 | 98 | 11.2 | 40.6 | 8.53 | 1.188 | 7.26 | 1.077 | 5.9 | 1.105 | 2.99 | 0.408 | 2.47 | 0.344 |
| C 14 | 41 | 87 | 9.89 | 36.2 | 7.78 | 0.982 | 6.68 | 0.99 | 5.31 | 0.969 | 2.62 | 0.353 | 2.09 | 0.289 |
| C 15 | 47 | 97 | 10.8 | 39.1 | 7.96 | 1.21 | 6.9 | 1.014 | 5.64 | 1.063 | 2.97 | 0.417 | 2.51 | 0.348 |
| C 16 | 44 | 91 | 10.2 | 37.4 | 7.82 | 1.156 | 6.74 | 0.997 | 5.62 | 1.048 | 2.84 | 0.387 | 2.31 | 0.323 |
| C 17 | 49 | 101 | 11.4 | 42 | 8.73 | 1.257 | 7.45 | 1.116 | 6.22 | 1.184 | 3.24 | 0.451 | 2.73 | 0.384 |
| C 18 | 38 | 79 | 8.55 | 31.9 | 6.74 | 1.268 | 6.11 | 0.918 | 5.28 | 1.017 | 2.86 | 0.392 | 2.39 | 0.34 |
| C 19 | 41 | 83 | 9.58 | 35.1 | 7.25 | 1.368 | 6.46 | 0.971 | 5.59 | 1.107 | 3.14 | 0.437 | 2.72 | 0.386 |
| C 19' | 41 | 80 | 9.34 | 34.4 | 7.08 | 1.33 | 6.29 | 0.955 | 5.53 | 1.086 | 3.08 | 0.431 | 2.66 | 0.384 |
| C 20 | 55 | 113 | 12.4 | 45.2 | 9.24 | 1.299 | 7.9 | 1.171 | 6.31 | 1.148 | 3.11 | 0.417 | 2.53 | 0.35 |
| DB3-6 | 15 | 35 | 3.76 | 13.9 | 3.3 | 0.477 | 2.92 | 0.464 | 2.43 | 0.414 | 1.04 | 0.129 | 0.7 | 0.091 |
| DB3-6' | 15 | 34 | 3.84 | 14.2 | 3.27 | 0.483 | 2.98 | 0.471 | 2.48 | 0.423 | 1.06 | 0.129 | 0.7 | 0.092 |
| DB3-8 | 21 | 48 | 5.36 | 19.2 | 4.48 | 0.505 | 3.9 | 0.591 | 2.97 | 0.474 | 1.12 | 0.131 | 0.68 | 0.087 |
| DB3-10 | 24 | 53 | 6.03 | 22.2 | 4.85 | 0.598 | 3.95 | 0.576 | 2.96 | 0.497 | 1.24 | 0.156 | 0.87 | 0.115 |
| DB3-12 | 15 | 35 | 3.95 | 14.2 | 3.42 | 0.492 | 2.97 | 0.453 | 2.19 | 0.343 | 0.8 | 0.094 | 0.5 | 0.065 |
| DB3-14 | 13 | 30 | 3.36 | 12.4 | 2.97 | 0.459 | 2.68 | 0.433 | 2.29 | 0.398 | 0.98 | 0.118 | 0.65 | 0.084 |
| DB3-16 | 15 | 34 | 3.81 | 13.8 | 3.28 | 0.516 | 2.9 | 0.448 | 2.22 | 0.349 | 0.8 | 0.093 | 0.5 | 0.062 |
| DB3-18 | 19 | 43 | 4.82 | 17.5 | 4.15 | 0.509 | 3.7 | 0.568 | 2.75 | 0.412 | 0.93 | 0.112 | 0.64 | 0.086 |
| DB-6 | 11 | 29 | 2.82 | 10.7 | 2.71 | 0.468 | 2.53 | 0.412 | 2.19 | 0.364 | 0.89 | 0.106 | 0.57 | 0.073 |
| DB-8 | 19 | 46 | 5.1 | 19.2 | 4.79 | 0.511 | 4.58 | 0.772 | 4.17 | 0.724 | 1.85 | 0.226 | 1.23 | 0.162 |
| DB-10 | 24 | 58 | 5.6 | 20.8 | 4.74 | 0.66 | 4.1 | 0.63 | 3.39 | 0.6 | 1.56 | 0.203 | 1.16 | 0.157 |
| DB-12 | 18 | 42 | 4.67 | 17.3 | 4.34 | 0.497 | 4.12 | 0.682 | 3.54 | 0.589 | 1.46 | 0.176 | 0.93 | 0.122 |
| DB-14 | 18 | 40 | 4.63 | 17.4 | 4.43 | 0.477 | 4.49 | 0.777 | 4.25 | 0.743 | 1.82 | 0.222 | 1.19 | 0.156 |
| DB-16 | 13 | 31 | 3.55 | 13.3 | 3.41 | 0.506 | 3.35 | 0.567 | 2.95 | 0.469 | 1.11 | 0.132 | 0.7 | 0.088 |
| DB-18 | 17 | 44 | 4.54 | 16.5 | 4.07 | 0.454 | 3.82 | 0.607 | 3 | 0.463 | 1.06 | 0.13 | 0.74 | 0.098 |
| DF-6 | 76 | 165 | 18.4 | 66.2 | 14.4 | 1.404 | 12.1 | 1.769 | 9.43 | 1.664 | 4.35 | 0.563 | 3.33 | 0.453 |

| Sample | | | | | | | | | | | | | |
|---|---|---|---|---|---|---|---|---|---|---|---|---|---|
| DF-6' | 80 | 165 | 18.4 | 67 | 14.4 | 1.429 | 12.3 | 1.816 | 9.48 | 1.684 | 4.44 | 0.566 | 3.31 | 0.457 |
| DF-8 | 73 | 164 | 17.5 | 63.7 | 14.4 | 1.08 | 12 | 1.737 | 8.31 | 1.299 | 3.08 | 0.364 | 1.92 | 0.246 |
| DF-10 | 70 | 154 | 17.3 | 63.6 | 13.4 | 1.28 | 10.9 | 1.549 | 8.04 | 1.374 | 3.6 | 0.466 | 2.69 | 0.36 |
| DF-12 | 100 | 217 | 25.2 | 88 | 20.2 | 1.216 | 17.5 | 2.619 | 13.1 | 2.16 | 5.29 | 0.631 | 3.39 | 0.453 |
| DF-14 | 68 | 152 | 17.1 | 61.8 | 13.9 | 0.942 | 11.6 | 1.692 | 7.9 | 1.205 | 2.76 | 0.31 | 1.67 | 0.213 |
| DF-16 | 84 | 184 | 21.8 | 74.9 | 17.2 | 1.094 | 14.8 | 2.255 | 11.6 | 1.931 | 4.72 | 0.577 | 3.2 | 0.411 |
| DF-18 | 85 | 197 | 21.4 | 73.3 | 16.4 | 1.451 | 13.9 | 2.011 | 9.72 | 1.541 | 3.72 | 0.469 | 2.71 | 0.371 |
| GeoPT28[M] | 51±1.4 | 106±3.2 | 12.1±0.2 | 45±1.2 | 9.1±0.21 | 1.84±0.03 | 7.8±0.08 | 1.15±0.019 | 6.5±0.16 | 1.3±0.02 | 3.7±0.05 | 0.53±0.01 | 3.38±0.07 | 0.49±0.009 |
| GeoPT28[C] | 53±0.6 | 108.2±0.92 | 12.6±0.1 | 49.2±0.5 | 9.62±0.11 | 1.98±0.018 | 8.54±0.1 | 1.23±0.016 | 7.1±0.089 | 1.36±0.017 | 3.79±0.05 | 0.56±0.01 | 3.64±0.039 | 0.54±0.008 |
| NIST-2704[M] | 30±1.2 | 59±1.9 | 7±0.08 | 27±0.35 | 5.5±0.04 | 1.2±0.01 | 5.3±0.04 | 0.82±0.01 | 5±0.03 | 1.02±0.01 | 2.96±0.04 | 0.43±0.005 | 2.72±0.03 | 0.39±0.0002 |
| NIST-2704[C] | 29 | 72 | 7.2 | 32 | 6.7 | 1.3 | 5.5 | 0.89 | 6 | 1.2 | 3.3 | 0.48 | 2.8 | 0.6 |
| GS-N[M] | 75±4.2 | 138±7.3 | 14±0.7 | 46±2.5 | 7.1±0.37 | 1.5±0.08 | 4.8±0.23 | 0.59±0.03 | 3±0.16 | 0.55±0.03 | 1.5±0.08 | 0.21±0.01 | 1.35±0.07 | 0.20±0.01 |
| GS-N[C] | 75±7.3 | 135±45.12 | 14.5 | 49±3.54 | 7.5±0.51 | 1.7±0.16 | 5.2±0.79 | 0.6±0.08 | 3.1±1.04 | 0.68±0.18 | 1.5±0.4 | 0.22 | 1.4±0.54 | 0.22±0.06 |

Note: Superscripts M and C correspond to average measured and consensus values of reference material mentioned in earlier tables.







**Table 4.** Re-Os isotopes composition and Re-Os abundance analyzed in selected cryoconite (C), moraine (DF:<63 μm), diesel engine exhaust (DEE) and Indian coal (GC: Gondwana coal, TC: Tertiary coal) samples. The terms LM (Local Moraine), LR (Less Radiogenic with low Os concentration) and UR (Ultramafic Rock) are the source end members contributing Os to the glacier samples.

| Sample | Total Re(2s) | Total Os(2s) | $^{192}$Os(2s) | $^{187}$Re/$^{188}$Os | $^{187}$Os/$^{188}$Os | rho | %Re | %$^{187}$Os | %$^{188}$Os | % Os source contribution(1s) | | |
|---|---|---|---|---|---|---|---|---|---|---|---|---|
| | (ng g$^{-1}$) | (pg g$^{-1}$) | (pg g$^{-1}$) | 2s | 2s | | Blank | Blank | Blank | LM | LR | UR |
| C1 | 0.206± 0.002 | 32.3± 0.2 | 12.1± 0.1 | 33.8± 0.4 | 0.911± 0.008 | 0.402 | 1.16 | 0.09 | 0.33 | 66.73 | 31.78 | 1.5 |
| C3 | 0.163± 0.002 | 10.9± 0.1 | 3.9± 0.1 | 83± 1.4 | 1.278± 0.170 | 0.599 | 1.47 | 0.2 | 1.03 | 30.69 | 69.16 | 0.15 |
| C5 | 0.211± 0.002 | 14.2± 0.1 | 5.1± 0.1 | 82.4± 1.1 | 1.310± 0.015 | 0.596 | 1.13 | 0.15 | 0.79 | 40.73 | 59.1 | 0.17 |
| C9 | 0.235± 0.002 | 37± 0.2 | 14.1± 0.1 | 33.3± 0.4 | 0.790± 0.007 | 0.414 | 1.02 | 0.09 | 0.29 | 70.48 | 27.95 | 1.57 |
| C11 | 0.252± 0.002 | 77.3± 0.3 | 30.5± 0.2 | 16.4± 0.2 | 0.479± 0.004 | 0.386 | 0.95 | 0.07 | 0.13 | 69.85 | 24.06 | 6.09 |
| C13 | 0.463± 0.002 | 63.6± 0.3 | 24.2± 0.1 | 38.1± 0.3 | 0.777± 0.006 | 0.522 | 0.52 | 0.05 | 0.17 | 92.35 | 4.58 | 3.07 |
| C14 | 0.416± 0.002 | 39.4± 0.2 | 14.9± 0.1 | 55.5± 0.5 | 0.829± 0.007 | 0.543 | 0.58 | 0.08 | 0.27 | 74.41 | 23.46 | 2.13 |
| C17 | 0.35± 0.002 | 41.1± 0.2 | 15.6± 0.1 | 44.7± 0.4 | 0.803± 0.006 | 0.504 | 0.69 | 0.08 | 0.26 | 74.59 | 23.11 | 2.3 |
| C18 | 0.403± 0.002 | 104± 0.4 | 41.5± 0.2 | 19.3± 0.2 | 0.378± 0.003 | 0.483 | 0.6 | 0.06 | 0.1 | 69.65 | 21.02 | 9.33 |
| C19 | 0.465± 0.002 | 55± 0.2 | 21.1± 0.1 | 43.9± 0.3 | 0.702± 0.006 | 0.531 | 0.51 | 0.07 | 0.19 | 84.81 | 11.67 | 3.53 |
| DF6 | 0.208± 0.002 | 30.4± 0.2 | 11.1± 0.1 | 37.4± 0.5 | 1.118± 0.010 | 0.408 | 1.15 | 0.08 | 0.36 | | | |
| DF12 | 0.141± 0.002 | 12.5± 0.1 | 4.4± 0.1 | 63.9± 1.2 | 1.478± 0.018 | 0.518 | 1.7 | 0.15 | 0.91 | | | |
| GC1 | 0.261± 0.002 | 6.9± 0.1 | 2.4± 0.1 | 217± 8.2 | 1.644± 0.069 | 0.856 | 4.39 | 0.83 | 6.38 | | | |
| GC2 | 0.757± 0.003 | 17.9± 0.1 | 6.2± 0.1 | 243± 4.2 | 1.606± 0.032 | 0.801 | 1.52 | 0.33 | 2.56 | | | |
| TC1 | 0.527± 0.003 | 726± 2.8 | 299± 2.6 | 3.5± 0.04 | 0.144± 0.002 | 0.609 | 2.18 | 0.08 | 0.05 | | | |
| TC2 | 0.468± 0.002 | 63.4± 0.3 | 25.9± 0.3 | 35.9± 0.4 | 0.213± 0.003 | 0.562 | 2.46 | 0.59 | 0.63 | | | |
| DEE1 | 0.073± 0.011 | 2.1± 0.5 | 0.9± 0.5 | 166± 95 | 0.218± 0.185 | 0.631 | 48.4 | 63.23 | 40.75 | | | |
| DEE2 | 1.032± 0.018 | 5.7± 0.8 | 2.3± 0.8 | 884± 294 | 0.205± 0.126 | 0.541 | 10.48 | 64.54 | 39.86 | | | |

5    Note: Total procedural blanks were 2.1 ppt for Re and 0.1 ppt for Os, with a $^{187}$Os/$^{188}$Os ratio of 0.25.