# Peer review of "Osmium isotope and trace elements reveal melting of Chhota Shigri Glacier, western Himalaya, insensitive to anthropogenic emission residues"

_The Cryosphere, 2020_

## Referee Comment (RC1) · Anonymous Referee #1 · 12 Sep 2020

This manuscript concerns the geochemical characterization of cryoconite from an Himalayan glacier (western Himalaya). The authors applied several techniques to this aim: elemental and isotopic ones. The novelty of this work is the application of Re/Os isotopic systematic in order to evaluate the contribution of anthropogenic atmospheric emissions with respect to cryoconite composition. Combining several pieces of information from major, REE, trace elements and isotopic signatures, they conclude that cryoconite from the Chhota Shigri Glacier has a typical crustal signature, with only secondary inputs from anthropogenic emissions.

[Figure]

I highlight my lack of competence to judge the methodological side of this work, in particular the one related to isotopic analyses. The authors seem quite confident and I have nothing to report, but I guess that an opinion from an isotopic geochemist would be desirable.

In general the manuscript is well written and easy to follow, but my impression is that this work, in its current form, would be more appropriate for a geochemistry journal. The Cryosphere should publish papers dealing with the diverse aspects of glaciology, including glacial geochemistry. With glacial geochemistry I intend the geochemical processes which are somehow related to glacial environment. In this paper the authors present a detailed geochemical characterization of cryoconite, but they don't really link their findings with glacial (or supra-glacial) processes. For this reason, I believe that this manuscript should be published in a journal more focused on geochemistry, where their detailed geochemical analyses would be more appreciated. Otherwise the authors could deeply revise the manuscript, trying to better link their findings with glacial processes and highlighting the novelty of their method to evaluate the anthropogenic influence on cryoconite composition.

In addition, they should also shorten the side related to geochemistry (and this is a pity since the data are good, but not very suited for this journal), taking into consideration the possibility not to present all their data, which are so many. For example figures 6-7-8 and the associated discussion could be removed, I don't' think that their removal would worsen the quality of the paper. Non-geochemists will have great difficulties to follow the paper if the manuscript would be published as it is now. If the authors want to publish their work in The Cryosphere, I suggest to them to select a limited dataset to present here (for example isotopic data and normalized elemental patterns) and focus their attention on the discussion about the anthropogenic influence on cryoconite composition.

The authors should also compare their results concerning elemental analyses with previous results (see for example Owen et al., 2019; Baccolo et al., 2017; Lokas et al.,

2016; Singh et al., 2013) and better investigate the fact that cryoconite from the Chhota Shigri Glacier seems quite pristine, while other studies focused on Himalayan glaciers and cryoconite showed that pollution is strongly present.

One thing which is not clear to me is the link throughout the entire manuscript of pollution, glacier mass balance, carbonaceous compounds. These are important topics, but this work does not deal with them, so I believe it would be more appropriate to remove them.

Considering the above, I cannot support the publication of this work in The Cryosphere in its current form.

More specific comments

Please improve the abstract, now it is the most difficult part of the manuscript to follow. It is not very explicative I guess that you could drastically shorten it. The important things to say are: 1- Himalayan glaciers are rich in supra-glacial debris, also linked with human activities; 2- you have geochemically characterized cryoconite samples from an Himalayan glacier, also applying cutting edge methods (osmium isotopes); 3-your results show that cryoconite on your glacier has a fully crustal signature, regardless the data you consider (major and trace elements, osmium isotopes); 4- provide some information about the scientific significance of such results.

Line 11-16: please rephrase, these sentences are very difficult to follow and not grammatically perfect.

Line 16: you introduce emission residues and then you suddenly turn to metals. Please better introduce metals as one of the most important anthropogenic species spread in the environment.

Line 20-21: "a benchmark glacier for process understanding in the western Himalaya" what does it mean?

Line 22: maybe change "composition" with "signature"?

Line 23: change "compositions" with "values"

Line 8-10: you write "Given that the presence of anthropogenic emission residues on the Himalaya is linked to driving climate change, enhanced glacier melting, and downstream water resources,...". I don't really agree with this passage. The most important process linked to the presence of anthropogenic species on glaciers is the increase of human atmospheric emissions, mostly related to industrial activities and transport. What is the link between climate change and anthropogenic pollution on glaciers? Please reformulate this passage. Also the role of glacier melting and downstream water resources is not clear in driving the presence of anthropogenic species on Himalayan glaciers.

Line 14-19: the authors state that scientific research on anthropogenic species found on Himalayan glaciers mostly focused on carbonaceous compounds and that studies dealing with metals are not common. This is not true! There are tens of papers showing that the concentration of many elements into glacier ice, in particular heavy metals, has increased in the last decades. If the authors look in scholar for the words "ice core asia metals", they will find heaps of interesting papers to cite.

Line 19-22: also this statement is not correct. For example look at Beaudon et al., 2018 (Central Tibetan Plateau atmospheric trace metals contamination: A 500-year record from the Puruogangri ice core), you will find that also for metals there are some works discussing their probable source.

Line 3-5: "given that Re-Os isotopes are independent to the rate and magnitude of emission, biological or physiochemical fractionation during transport, complex orography and meteorological parameters." I would simplify as follows: "given that the Re-Os

isotopic signature is strongly conservative. It mostly depends on the emission source of the considered material and for this reason it is adopted in provenance studies (add a ref about this point)."

Line 23: "and is therefore an excellent site to study long-distance emission inputs." Why? I don't get the point, please explain better

Line 25: "4050 and 6263"

Line 25-26: "a benchmark glacier for process understanding in the western Himalaya" also this passage is not clear. Why is it considered a benchmark glacier? What processes are you talking about?

Line 4: "comprises"

Line 15: "sample powder"

Line 26: "sample duplicates"

Line18: "are common"

References:

Baccolo et al., 2017. Cryoconite as a temporary sink for anthropogenic species stored in glaciers Beaudon et al., 2018. Central Tibetan Plateau atmospheric trace metals contamination: A 500-year record from the Puruogangri ice core Lokas et al., 2016. Accumulation of atmospheric radionuclides and heavy metals in cryoconite holes on an Arctic glacier Owens et al. 2019. Extreme levels of fallout radionuclides and other contaminants in glacial sediment (cryoconite) and implications for downstream aquatic ecosystems Singh et al. 2013. Atmospheric deposition studies of heavy metals in

Arctic by comparative analysis of lichens and cryoconite

best regards

---

## Referee Comment (RC2) · Anonymous Referee #2 · 27 Nov 2020

First, I am very sorry that this review is so late. Second, I am even more sorry that my review is so negative.

The introduction to the paper is poorly focused. The paper is trying to ascertain the level of anthropogenic pollution in the catchment via an analysis of cryoconite, moraine, coal and diesel exhaust. The conclusion is that there is little evidence of pollution, and I think that this may be because the sampling and methodology may be not fit for purpose. I could be very wrong, but the text does not provide a convincing argument that the sampling and methodology is fit to deliver the aims and objectives of the paper.

[Figure]

First, which pollutants are you most interested in and why have you selected these?

Second, how will you know if these pollutants are above naturally occurring levels on your glacier?

Third, why should cryoconite be polluted relative to the local rock you sample? What assumptions do you make and have you tested any of them?

Fourth, how much variability is there in the pollutant content of coal and diesel? Are your samples representative of the pollutants being deposited on the glacier from these sources?

Fifth, why aren't you using sequential extraction, rather than analysis of whole rocks only, to look for evidence of trace metal pollution. Cryoconite has a large mineral component from melt out debris and local aerosol. Don't you have a problem of signal to background to contend with? Whole rock digestion will bias your results to a comparison of these minerals to the rock samples you collected from the moraine.

Finally, you need to be very clear about why and for what purpose you are using the isotopes you employ. It is unclear from the introduction why do include these data.

Where I am is that I am unsure about the conclusions you reach, given the lack of justification of the methodology and sampling you have undertaken.

---

## Author Comment (AC1) · 18 Dec 2020

Note: RC and AR correspond to Referee's Comment and Author's Response respectively

RC: This manuscript concerns the geochemical characterization of cryoconite from a Himalayan glacier (western Himalaya). The authors applied several techniques to this aim: elemental and isotopic ones. The novelty of this work is the application of Re/Os isotopic systematic in order to evaluate the contribution of anthropogenic atmospheric

emissions with respect to cryoconite composition. Combining several pieces of information from major, REE, trace elements and isotopic signatures, they conclude that cryoconite from the Chhota Shigri Glacier has a typical crustal signature, with only secondary inputs from anthropogenic emissions. I highlight my lack of competence to judge the methodological side of this work, in particular the one related to isotopic analyses. The authors seem quite confident and I have nothing to report, but I guess that an opinion from an isotopic geochemist would be desirable.

AR: Thanks for appreciating the methodology of the paper.

RC: In general, the manuscript is well written and easy to follow, but my impression is that this work, in its current form, would be more appropriate for a geochemistry journal. The Cryosphere should publish papers dealing with the diverse aspects of glaciology, including glacial geochemistry. With glacial geochemistry, I intend the geochemical processes which are somehow related to glacial environment. In this paper the authors present a detailed geochemical characterization of cryoconite, but they don't really link their findings with glacial (or supra-glacial) processes. For this reason, I believe that this manuscript should be published in a journal more focused on geochemistry, where their detailed geochemical analyses would be more appreciated. Otherwise the authors could deeply revise the manuscript, trying to better link their findings with glacial processes and highlighting the novelty of their method to evaluate the anthropogenic influence on cryoconite composition.

AR: Thanks for appreciating our writing style, and we are glad that the paper was easy to follow, even to a non-isotope geochemist. We understand the referee's concern. But we would like to emphasize again that our study for the first time demonstrates that dark-colored materials, in our case cryoconite, deposited on the surface of Chhota Shigri Glacier (CSG) in the western Himalaya is derived from natural sources. The particles are mostly locally sourced, with minor long-range inputs from the Thar and the Sahara Desert. This contrasts with many recent findings that show substantial anthropogenic input (e.g., Li et al., 2016, Sources of black carbon to the Himalayan–Tibetan

[Figure]

Plateau glaciers, Nature Communication). In our opinion this is a significant finding, as we show that the surface of CSG is essentially free of anthropogenically emitted particles, but yet, CSG is losing glacial mass at an average rate of 0.50 meter water-equivalent per year (0.43 m w.e. yr-1) over the last two decades, which is higher than the central (0.35 m w.e.ye-1) and eastern Himalayan (0.43 m w.e. yr-1) glaciers that receive a substantial contribution of anthropogenic sources generated from the Indian subcontinent. We would like to highlight that CSG is considered to be a benchmark glacier in the western Himalaya, and it is one of the cleanest in terms of debris cover and glacial impurities. Our study proves glacial mass wastage rates in CSG is insensitive to glacial impurities. Therefore, we strongly feel that our study would be much more appreciated in the glaciological community as using state-of-the-art geochemical tools (first Os isotope dataset in the high-altitude Himalaya) we ultimately decipher the glacial environment. The study would lose its audience and impact if we submit the paper in a Geochemistry based journal.

RC: In addition, they should also shorten the side related to geochemistry (and this is a pity since the data are good, but not very suited for this journal), taking into consideration the possibility not to present all their data, which are so many. For example, figures 6-7-8 and the associated discussion could be removed, I don't' think that their removal would worsen the quality of the paper. Non-geochemists will have great difficulties to follow the paper if the manuscript would be published as it is now. If the authors want to publish their work in The Cryosphere, I suggest to them to select a limited dataset to present here (for example isotopic data and normalized elemental patterns) and focus their attention on the discussion about the anthropogenic influence on cryoconite composition.

AR: This is a very good suggestion. A lot of material (including figures 6, 7, 8) can be easily removed from the main text. We agree that removing these figures and their associated write-up would not impact by any means worsen the quality of the paper. Therefore, if given a chance, the manuscript can be shortened only by highlighting the

anthropogenic influence on cryoconite composition.

RC: The authors should also compare their results concerning elemental analyses with previous results (see for example Owen et al., 2019; Baccolo et al., 2017; Lokas et al., 2016; Singh et al., 2013) and better investigate the fact that cryoconite from the Chhota Shigri Glacier seems quite pristine, while other studies focused on Himalayan glaciers and cryoconite showed that pollution is strongly present.

AR: Precisely, that is what we would like to highlight, we already stated this with Li et al. paper (e.g., Li et al., 2016, Sources of black carbon to the Himalayan–Tibetan Plateau glaciers, Nature Communication) study. Thanks for pointing us toward new references that would further support our claim, and highlight the importance of this study.

RC: One thing which is not clear to me is the link throughout the entire manuscript of pollution, glacier mass balance, carbonaceous compounds. These are important topics, but this work does not deal with them, so I believe it would be more appropriate to remove them. Considering the above, I cannot support the publication of this work in The Cryosphere in its current form.

AR: We would revise as per the referee's comments. Here we would like to highlight that a separate independent study was carried out by our group on the same set of cryoconite samples. In the Nizam et al., 2020 paper using the distribution of organic carbon (OC) activation energy and 14C activity we demonstrated that $98.3 \pm 1.6\%$ and $1.7 \pm 1.6\%$ of the OC in the cryoconite samples studied here are derived from biomass and petrogenic sources, respectively. As cryoconite is a mixture of dust, soot, and microbes, this and the Nizam et al., 2020 study completes all the geochemical spectrum of CSG cryoconite including organic and inorganic impurities. We would like to highlight the Nizam et al., 2020 paper was published as a cover page article in Environmental Science and Technology. So, we do have some link with the carbonaceous compounds, but we agree, that we do not quantitatively deal with mass balance issues. We would be happy to introduce better clarity in the manuscript with addition/omission

of text, as and when required.

Specific comments: RC: Please improve the abstract, now it is the most difficult part of the manuscript to follow. It is not very explicative I guess that you could drastically shorten it. The important things to say are: 1- Himalayan glaciers are rich in supraglacial debris, also linked with human activities; 2- you have geochemically characterized cryoconite samples from a Himalayan glacier, also applying cutting edge methods (osmium isotopes); 3-your results show that cryoconite on your glacier has a fully crustal signature, regardless the data you consider (major and trace elements, osmium isotopes); 4- provide some information about the scientific significance of such results.

AR: We agree that the abstract could have been shortened. Please see the revised abstract below: The western Himalaya and the Karakoram region hold ca. 70% of the total ice volume of the Himalaya that seasonally melts which is, in part, controlled by the presence of supra-glacial debris. However, the source, origin, and pathways of this supra-glacial debris on the ice surface of Himalayan glaciers remain poorly constrained. Here, we present major and trace element geochemistry, rhenium-osmium ($187Os/188Os$) isotopes composition of cryoconite: a dark-colored aggregate of mineral and organic materials-on the ablation zone (4100-4900 m a.s.l.) of the Chhota Shigri Glacier (CSG) in the western Himalaya. Using multiple lines of geochemical evidence, we show that the surface of CSG is essentially free of anthropogenically emitted particles, contrary to many previous findings. Given that CSG has limited debris cover (ca. 3.4%) and the presence of anthropogenically derived particles were not appreciably detected, we conclude that accelerated mass loss in the CSG is considered to be primarily related to the increase of the Earth's near-surface temperature in direct response to global warming.

Page 1 RC: Line 11-16: please rephrase, these sentences are very difficult to follow and not grammatically perfect.

AR: Thanks for the suggestion. We will rephrase it in the revised manuscript.

RC: Line 16: you introduce emission residues and then you suddenly turn to metals. Please better introduce metals as one of the most important anthropogenic species spread in the environment.

AR: Noted, and will be revised.

RC: Line 20-21: "a benchmark glacier for process understanding in the western Himalaya" what does it mean?

AR: Chhota Shigri Glacier (CSG) is considered to be a benchmark glacier in the western Himalaya (Wagon et al., 2007; Pandey et al., 2017) as (i) it is medium-sized (15.7 km2 over a 34.7 km2 catchment (ii) limited debris cover ($\sim$ 3.4%) (iii) lies in the crucial area alternately influenced by the southwest monsoon in summer and by the westerlies in winter and (iv) extensively studied over the last 2 decades. So, CSG is considered as a benchmark glacier by the Indian and the International community. We would explicitly state this in the revised manuscript.

RC: Line 22: maybe change "composition" with "signature"?

AR: Noted, will be corrected in revised draft.

RC: Line 23: change "compositions" with "values"

AR: Noted, will be corrected in revised draft.

Page 2 RC: Line 8-10: you write "Given that the presence of anthropogenic emission residues on the Himalaya is linked to driving climate change, enhanced glacier melting, and downstream water resources,.. .". I don't really agree with this passage. The most important process linked to the presence of anthropogenic species on glaciers is the increase of human atmospheric emissions, mostly related to industrial activities and transport. What is the link between climate change and anthropogenic pollution on glaciers? Please reformulate this passage. Also, the role of glacier melting and downstream water resources is not clear in driving the presence of anthropogenic species on Himalayan glaciers.

AR: We agree with the referee's point and will revise the text. For example, "Presence of anthropogenic emission residues on the glacial surface has been linked to enhanced glacier melting..." The paragraph will be rephrased, removing the ambiguous term 'downstream water'.

RC: Line 14-19: the authors state that scientific research on anthropogenic species found on Himalayan glaciers mostly focused on carbonaceous compounds and that studies dealing with metals are not common. This is not true! There are tens of papers showing that the concentration of many elements into glacier ice, in particular heavy metals, has increased in the last decades. If the authors look in scholar for the words "ice core asia metals", they will find heaps of interesting papers to cite.

AR: The referee's argument is true, but available studies are mainly restricted to the central/eastern Himalaya and Tibetan regions. But we understand his/her concern and will downplay our statement accordingly. We would like to highlight that the rhenium-osmium (187Os/188Os) isotopic data-that are widely used proxy to track contribution from catalytic converters fitted in automobile exhausts—are new and probably the first dataset over the Himalayan glaciers.

RC: Line 19-22: also this statement is not correct. For example, look at Beaudon et al., 2018 (Central Tibetan Plateau atmospheric trace metals contamination: A 500-year record from the Puruogangri ice core), you will find that also for metals there are some works discussing their probable source.

AR: We will keep this in mind when revising the manuscript.

Page 3 RC: Line 3-5: "given that Re-Os isotopes are independent to the rate and magnitude of emission, biological or physiochemical fractionation during transport, complex orography, and meteorological parameters." I would simplify as follows: "given that the Re-Os isotopic signature is strongly conservative. It mostly depends on the emission source of the considered material and for this reason, it is adopted in provenance studies (add a ref about this point)."

[Figure]

AR: Noted, and will be revised.

RC: Line 23: "and is therefore an excellent site to study long-distance emission inputs." Why? I don't get the point, please explain better.

AR: Chhota Shigri glacier has the lowest debris coverage in its ablation zone (ca. 3.4%), it is one of the cleanest glaciers (in relation to debris cover). Therefore, chances that the long-range transported anthropogenic particles will be diluted with an overwhelming local signature will be minimal. Thus, we stated that it is an excellent site to study "long-distance emission inputs". We would clarify this in the revised text.

RC: Line 25: "4050 and 6263"

AR: Noted, and will be revised

RC: Line 25-26: "a benchmark glacier for process understanding in the western Himalaya" also this passage is not clear. Why is it considered a benchmark glacier? What processes are you talking about?

AR: Chhota Shigri Glacier (CSG) is considered to be a benchmark glacier in the western Himalaya (Wagon et al., 2007; Pandey et al., 2017) as (i) it is medium-sized (15.7 km2 over a 34.7 km2 catchment (ii) limited debris cover ($\sim$ 3.4%) (iii) lies in the crucial area alternately influenced by the southwest monsoon in summer and by the westerlies in winter and (iv) extensively studied over the last 2 decades. So, CSG is considered as a benchmark glacier by the Indian and the International community. We would explicitly state this in the revised manuscript.

Page 4 RC: Line 4: "comprises"

AR: Noted, and will be revised

Page 6 RC: Line 15: "sample powder"

AR: Noted, and will be revised

RC: Line 26: "sample duplicates"

AR: Noted, and will be revised

Page 11 RC: Line18: "are common"

AR: Noted, and will be revised.

We hope that after these responses, the manuscript we will be allowed to carry out a detailed revision of the manuscript, followed by submission in The Cryosphere.

References Li et al. 2016. Sources of black carbon to the Himalayan–Tibetan Plateau glaciers. Nat. Commun. 7, 12574. Nizam et al. 2020. Biomass-derived provenance dominates glacial surface organic carbon in the western Himalaya. Environ. Sci. Technol., 54(14), 8612–8621. Pandey, et al. 2017. Regional representation of glaciers in Chandra Basin region, western Himalaya, India, Geosci. Front., 8(4), 841–850. Wagnon et al. 2009. Four years of mass balance on Chhota Shigri Glacier, Himachal Pradesh, India, a new benchmark glacier in the western Himalaya, J. Glaciol., 53 (183), 603–611.

Please also note the supplement to this comment:
https://tc.copernicus.org/preprints/tc-2020-165/tc-2020-165-AC1-supplement.pdf

---

## Author Comment (AC2) · 18 Dec 2020

Note: RC and AR correspond to Referee's Comment and Author's Response respectively

RC: First, I am very sorry that this review is so late. Second, I am even more sorry that my review is so negative. The introduction to the paper is poorly focused. The paper is trying to ascertain the level of anthropogenic pollution in the catchment via an analysis of cryoconite, moraine, coal and diesel exhaust. The conclusion is that there

is little evidence of pollution, and I think that this may be because the sampling and methodology maybe not fit for purpose. I could be very wrong, but the text does not provide a convincing argument that the sampling and methodology is fit to deliver the aims and objectives of the paper.

AR: We would try to revise the introduction of the paper if required. We would like to highlight that referee 1 praised our writing style. He/she writes "In general, the manuscript is well written and easy to follow". However, if the Editor feels that it needs to re-written, we would be happy to revise the introduction accordingly. The sampling and methodology part is written explicitly, and in fact, Referee 1 states "The authors seem quite confident and I have nothing to report on the methodology". I think the referee's questions the use of cryoconite to deliver the aims and objective of the paper. We would like to highlight that there are numerous papers (e.g. Owen et al., 2019; Baccolo et al., 2017; Lokas et al., 2016; Singh et al., 2013) that have used cryoconite (a mixture of dust, soot, and microbes) and achieved similar goals. For example, algae in cryoconite holes and mats on the ice surface can significantly decrease the albedo compared to clean ice (Figure 6, Yallop et al., 2012). Therefore, many factors are contributing to reduced albedo, and in turn melting, yet little is known about the interplay between dust, soot, and biology in the ablation zone. So, we don't see why the referee is not convinced. Nothing is new in the methodology, all aspect of the study from sampling and analytical is very well established.

RC: First, which pollutants are you most interested in and why have you selected these?

AR: As we stated, we are interested in metal impurities that get deposited along with other emission impurities including organic carbon (Sarwar et al., 2020). As stated earlier, the idea of my dissertation work was to characterize organic and inorganic impurities that are present in the CSG cryoconite. In this study, we only focused on inorganic (metal) impurities, whereas, Sarwar et al., 2020 reported organic impurities. We would like to highlight that each individual metal species comes from a specific

anthropogenic source or a mixture of many sources. For example, Os is mostly derived from catalytic converters, whereas Cd is derived from coal. All these metal impurities can be tagged to a specific anthropogenic source. However, our intention was not to identify the contribution of individual anthropogenic sources, but to look into the overall presence of anthropogenic particles, and then dive into possible sources. We selected these metals as they are widespread anthropogenic pollutants, and not to track their individual biogeochemical cycling.

RC: Second, how will you know if these pollutants are above naturally occurring levels on your glacier?

AR: This was already included in the manuscript, and perhaps referee 2 overlooked Figure 4. Trace metal enrichment exceeding the local background value (see Figure 4 in the original manuscript) provide the first-order evidence for the presence of any anthropogenic pollution in any pristine environment. Enrichment factor calculation (Figure 4) is a routine standard procedure to discriminate between natural and anthropogenic (metal) pollutants. To be more confident, we also used osmium isotopes ($187Os/188Os$) to quantify anthropogenic sources. Please see Figure 11a and its associated discussion in Section 4.2. As such we consider the presentation of the submitted paper already covers the referee's question.

RC: Third, why should cryoconite be polluted relative to the local rock you sample? What assumptions do you make and have you tested any of them?

AR: By definition, cryoconite is a mixture of dust, soot (anthropogenic particles), and microbes and they are regarded as a sink for the natural and anthropogenic sources in a glaciated environment (Baccolo et al., 2017, Lokas et al., 2016, Owen et al., 2019, Sing et al., 2013). Since anthropogenic sources have a very high amount of metal concentration, even a small contribution from such a source could be visible in the cryoconite, unless the signal is completely overwhelmed by the local signature. Moreover, $187Os/188Os$ crustal signature is around 1.4 whereas anthropogenic signature can be

as low as 0.16. We would like to emphasize this variability is huge, and to an isotope geochemist, even a third decimal place variability in the Os isotope is acceptable. We looked into multiple geochemical evidences such as inter-elemental relationship (e.g., Figures 6, 7, 8, 12) and in the osmium isotope space (Figures 10, 11a). All these data conclusively show the absence of the anthropogenic signal in any of the cryoconite sediments. Based on all the geochemical evidence we have concluded that the CSG surface is essentially free of anthropogenic particles. Our claim is further supported by our one more recent paper (Sarwar et al., 2020) where we looked into 14C activity and activation energy of OC. As such we consider the presentation of the submitted paper already covers the referees question, particularly given that the entire paper discusses multiple geochemical evidence, modeling, and quantitative assessments.

RC: Fourth, how much variability is there in the pollutant content of coal and diesel? Are your samples representative of the pollutants being deposited on the glacier from these sources?

AR: This is a good question. Cryoconite does not represent snow and ice—the main components of glaciers. So of course the obvious question is why not just measure the metal and its isotopic composition (e.g., 187Os/188Os) of snow and ice. While it will be truly ideal to know ice and snow composition, these analyses are extremely logistically challenging in this environment. One could melt ice and snow, further collect the material on a quartz filter (e.g. Wientjes et al., 2012), or bring back 10s of a liter of melted ice and snow meltwater back to the laboratory. Also, once we filter ice and snowmelt water, the anthropogenic particles can be much smaller than the pore size of a typical quartz filter and there will be a chance of chemical fractionation. Therefore, for the time being, we carried out the study in cryoconite debris, as our objective was to find out the presence and absence of anthropogenic particles in CSG, and these samples very well suit our objective. Coming back to the other question about concentration variability in coal and diesel. At this moment, we cannot assess the pollutant content variability of coal and diesel in CSG since these particles were absent in CSG. Our intention

was to only use coal and diesel compositional range as end-member compositions to deconvolute source end-member of Os using the three-component isotope mixing model. However, since all geochemical evidence pointed towards the absence of any fossil fuel source, we did not use coal and diesel compositions as source end-member composition. Moreover, we also had ancillary datasets on the distribution of OC activation energy, 14C activity, and radiogenic isotopes of 208Pb/204Pb, 207Pb/204Pb, and 206Pb/204Pb in these samples that conclusively demonstrate that 98.3± 1.6% and 1.7± 1.6% of the OC in western Himalayan glaciers are derived from biomass and petrogenic sources, respectively, and there is negligible fossil fuel component. Therefore, based on all organic and inorganic tracers we concluded that there is no fossil fuel signature in CSG. We agree that there will be a lot of chemical heterogeneity in the coal and diesel samples, but, since we did not use them in the mixing model or talked about spatial variability, we can negate the concern.

RC: Fifth, why aren't you using sequential extraction, rather than analysis of whole rocks only, to look for evidence of trace metal pollution. Cryoconite has a large mineral component from melt out debris and local aerosol. Don't you have a problem of signal to the background to contend with? Whole-rock digestion will bias your results to a comparison of these minerals to the rock samples you collected from the moraine.

AR: This is a very good question and we understand the concern. Again, we would like to iterate that measuring 187Os/188Os on leachates is analytically very challenging. Therefore, we relied on fine fractions (<63$\mu$m) with the assumption that would contain long-range transported Os particles. Wherever it was possible, we did sequential extraction. For example, on the same set of samples, we have used a sequential extraction procedure (RPO analysis), which involves continuous sample heating, which leads to the release of $CO_2$. Each fraction was cryogenically purified, trapped and flame sealed into a glass tube for detailed C-isotope analysis. The $CO_2$ concentration in the carrier gas was continuously measured at a resolution of 1 second by an infrared gas analyzer. Each RPO fraction collected is graphitized with the radiocarbon abundance determined via Accelerated Mass Spectrometry whereas a 10% split of each RPO fraction was used for stable isotopes ($\delta$13C) analysis using a dual-inlet Isotope Ratio Mass Spectrometer (IRMS). A detailed description of the sequential extraction procedure can be found in Sarwar et al., (2020). The sequential extraction procedure (RPO technique) also revealed the absence of anthropogenic particles, namely fossil fuel sources. Thus we acknowledge the concern, but, based on all geochemical evidence, we are certain that our reached conclusion is robust.

RC: Finally, you need to be very clear about why and for what purpose you are using the isotopes you employ. It is unclear from the introduction why do include these data. Where I am is that I am unsure about the conclusions you reach, given the lack of justification of the methodology and sampling you have undertaken.

AR: We do not agree with the referee's comment since we explicitly mentioned the scope of the osmium isotopes in source identification in environmental studies and why we selected it (Page# 3 lines 1-13 in the original manuscript). Similarly, we have also discussed the significance of different size fractions of local moraine in the methodology section to fix the local crustal and windblown signal (Page# 5 lines 24-26 original manuscript). What perhaps needs to be added in the methodology section is about the selection of coal and engine exhaust samples. We will therefore add a line in section 2.4.2 as "coal and diesel engine exhaust samples were selected to represent anthropogenic emission source end-member." As for as the referee's concern about our conclusion, we would like to mention that our outcome is simply based on the absence of anthropogenic signals. Given that there is no or limited anthropogenic emission residues, therefore, the glacier melting behavior will not be likely affected by anthropogenic emission residues as reported in other parts of the Himalaya and globe. We hope that after these responses, the manuscript we will be allowed to carry out a detailed revision of the manuscript, followed by submission in The Cryosphere.

References Baccolo et al. 2017. Cryoconite as a temporary sink for anthropogenic species stored in glaciers. Sci Rep., 7, 9623. Łokas et al. 2016. Accumulation

of atmospheric radionuclides and heavy metals in cryoconite holes on an Arctic glacier. Chemosphere, 160, 162-72. Owens et al. 2019. Extreme levels of fallout radionuclides and other contaminants in glacial sediment (cryoconite) and implications for downstream aquatic ecosystems. Sci Rep. 9, 12531. Singh et al. 2013. Atmospheric deposition studies of heavy metals in Arctic by comparative analysis of lichens and cryoconite. Environ. Monit. Assess., 185, 1367–1376. Wientjes et al. 2012. Carbonaceous particles reveal that Late Holocene dust causes the dark region in the western ablation zone of the Greenland ice sheet. J. Glaciol. 58(210) ,787-794. Yallop et al. 2012. Photophysiology and albedo-changing potential of the ice algal community on the surface of the Greenland ice sheet. ISME J, 6, 2302–2313.

Please also note the supplement to this comment:
https://tc.copernicus.org/preprints/tc-2020-165/tc-2020-165-AC2-supplement.pdf